# Prefrontal parvalbumin interneurons require juvenile social experience to establish adult social behavior

Lucy K. Bicks [1,2,3,4,5], Kazuhiko Yamamuro[1,2,3,4,5], Meghan E. Flanigan[2,5], Julia Minjung Kim[1,2,3,4,5], Daisuke Kato[1,2,3,4,5], Elizabeth K. Lucas [2,5], Hiroyuki Koike[1,2,3,4,5], Michelle S. Peng[1,2,3,4,5], Daniel M. Brady [1], Sandhya Chandrasekaran[1,2,5], Kevin J. Norman[1,2,3,4,5], Milo R. Smith[1,2,3,4,5], Roger L. Clem[2,5], Scott J. Russo [2,5], Schahram Akbarian [1,2,5,6] & Hirofumi Morishita [1,2,3,4,5,6✉]

Social isolation during the juvenile critical window is detrimental to proper functioning of the prefrontal cortex (PFC) and establishment of appropriate adult social behaviors. However, the specific circuits that undergo social experience-dependent maturation to regulate social behavior are poorly understood. We identify a specific activation pattern of parvalbumin-positive interneurons (PVIs) in dorsal-medial PFC (dmPFC) prior to an active bout, or a bout initiated by the focal mouse, but not during a passive bout when mice are explored by a stimulus mouse. Optogenetic and chemogenetic manipulation reveals that brief dmPFC-PVI activation triggers an active social approach to promote sociability. Juvenile social isolation decouples dmPFC-PVI activation from subsequent active social approach by freezing the functional maturation process of dmPFC-PVIs during the juvenile-to-adult transition. Chemogenetic activation of dmPFC-PVI activity in the adult animal mitigates juvenile isolation-induced social deficits. Therefore, social experience-dependent maturation of dmPFC-PVI is linked to long-term impacts on social behavior.

[1] Department of Psychiatry, Icahn School of Medicine at Mount Sinai, One Gustave L. Levy Place, New York, NY 10029, USA. [2] Department of Neuroscience, Icahn School of Medicine at Mount Sinai, One Gustave L. Levy Place, New York, NY 10029, USA. [3] Department of Ophthalmology, Icahn School of Medicine at Mount Sinai, One Gustave L. Levy Place, New York, NY 10029, USA. [4] Mindich Child Health and Development Institute, Icahn School of Medicine at Mount Sinai, One Gustave L. Levy Place, New York, NY 10029, USA. [5] Friedman Brain Institute, Icahn School of Medicine at Mount Sinai, One Gustave L. Levy Place, New York, NY 10029, USA. [6]These authors jointly supervised this work: Schahram Akbarian, Hirofumi Morishita. ✉email: hirofumi.morishita@mssm.edu

Social behavior is a fundamental process across species and encompasses a range of behaviors related to reproduction and parenting, as well as a host of complex social interactions outside of the mating context. Social experience during developmental windows is essential for the establishment of adult behavior. For example, in humans, institutionalized children who are placed into foster care during early development show improved cognitive outcomes compared with those who were never placed into foster care, or those placed into foster care at a later age[1,2]. In mice, social isolation during a specific juvenile window (p21–35) leads to long-lasting behavioral disruptions, implicating this time window as a key period for social experience-dependent development[3,4]. However, few studies have examined the cellular mechanisms regulating underlying experience-dependent changes supporting complex behavior development, such as social behavior. Given that social behavior deficits are a common dimension of many psychiatric disorders, identifying the specific circuits sensitive to experience-dependent modulation will likely point toward therapeutic targets that propel the amelioration of social-processing deficits shared across a range of disorders.

Evidence from human fMRI and circuit-based studies in rodents broadly implicates the evolutionarily conserved medial prefrontal cortex (mPFC) as part of a network that regulates social behavior[5]. In particular, the dorsal part of the mPFC, including the anterior cingulate cortex and the prelimbic region (dmPFC), has been particularly linked to social behavior[6,7]. During development, the PFC shows a protracted trajectory, maturing well into the juvenile period[8,9]. Juvenile PFC development is characterized by strengthening of inhibitory neurotransmission within the brain, which alters the excitatory and inhibitory balance[10,11]. Inhibitory GABAergic interneurons are a diverse population, made up of several sub-types with distinct connectivity, morphology, and physiology[12]. One such sub-type, molecularly defined by the expression of the calcium-binding protein parvalbumin (PV), shows an extended period of development[10,13] and is a known regulator of experience-dependent cortical development in primary sensory cortex[14]. However, the role of parvalbumin interneurons (PVIs) in juvenile social experience-dependent development is unknown. Both impairments in PVI functioning, particularly in the dmPFC, and decreased gamma oscillation power, indicative of decreased PVI activity, are common findings in neurodevelopmental disorders with social deficits[15–19], highlighting their potential role in social behavior development.

While much evidence implicates dmPFC-PVI dysfunction in humans and in animal models of neurodevelopmental disorders with social deficits[5], few studies have directly interrogated how social experience during development regulates PVI maturation and adult behavioral function. While recent studies demonstrated that continuous optogenetic or chemogenetic activation of mPFC-PVIs in adulthood can mitigate social deficits[20–22], the causal contribution of endogenous dmPFC-PVI activity to social behavior is not known. Importantly, social behavior is not one entity and is in fact composed of a range of highly specific behaviors. To establish dmPFC-PVIs as circuit targets for amelioration of social behavior deficits, it is essential to identify specific aspects of social behavior coupled with dmPFC-PVIs activity, and to assess whether dmPFC-PVIs undergo experience-dependent maturation to govern adult social behavior. Here, we show dmPFC-PVI development is affected by social experience during a juvenile critical window, resulting in disrupted activity patterns of dmPFC-PVIs and subsequent social behavior deficits.

## Results

**dmPFC-PVIs are activated prior to an active social bout.** As social behavior is composed of a range of highly specific behaviors, we first set out to examine to what extent adult dmPFC-PVIs are recruited during specific aspects of social behaviors. We used fiber photometry to monitor activity of dmPFC PVIs during social and object exploration in freely behaving mice[23]. We injected a Cre-dependent adeno-associated virus (AAV) encoding the fluorescence indicator of $Ca^{2+}$ signal, GCaMP6f, into the dmPFC of *PV-Cre* mice and implanted an optical fiber in the dmPFC in order to measure fluctuations of PVI activity during a social exploration task in adults (Fig. 1a, b; Supplementary Fig. 1). We recorded baseline PVI activity in an open-field (arena), and then added, in a counterbalanced fashion, a novel object (object) or a novel age/sex/strain-matched mouse (social). We found that dmPFC-PVI activity was significantly increased after introducing a novel animal, but not a novel object (Fig. 1c–e, Wilcoxon signed-rank test, social: $p = 0.03$, object: $p = 0.25$). These data demonstrate a rapid and preferential engagement of PVIs in the dmPFC during a naturalistic social encounter.

However, these findings are agnostic to the specific types of social encounters taking place during the social interaction, which are quite varied. We thus scored eight distinct and frequent social behaviors and grouped these behavioral categories into active, passive, or orient (see "Methods" for details). Behavioral epochs initiated by the focal mouse were termed "active bouts" and were defined as bouts that contained sub-behaviors that fell within the active behavior category that were sequential in time, with less than three seconds between the end of one and the start of another. The start of an active bout was defined as the beginning of the first of the behaviors within the active sequence (i.e orienting to the stimulus mouse if followed by an approach, see "Methods" for detail). Bouts initiated by the stimulus mouse were termed "passive bouts", and those during which the focal mouse was oriented toward the stimulus mouse without engaging in a subsequent interaction were termed "Orient bouts". We compared dmPFC-PVI activity occurring 3 s before and after the initiation of active and passive bouts (Fig. 1f–h). We found distinct dmPFC-PVI activity in these two bout types: immediately preceding the first active bout, we saw a brief increase in dmPFC-PVI activity, which was not sustained during the bout (Fig. 1f–h, left: one-way repeated measures ANOVA comparing baseline ($-3$ to $-2$ s), pre ($-1$ to $0$ s), and post ($0$ to $1$ s), $p = 0.02$, Tukey's test between baseline and pre, $p < 0.05$). These changes were not observed in the first passive bout nor were any changes observed prior to the first active object exploration bout (Fig. 1f–h, right: one-way repeated measures ANOVA comparing baseline, pre, and post, $p = 0.77$; Supplementary Fig. 2).

Collectively, our findings demonstrate a specific activation pattern of dmPFC PVIs only prior to an active bout, but not during a passive bout, suggesting a link between short dmPFC-PVI activation and subsequent bout initiation.

**Brief dmPFC-PVI activation promotes social approach.** In order to assess the causal relationship between brief dmPFC-PVI activation and subsequent active social behavior initiation, we set out to manipulate dmPFC-PVI activity on a fine timescale through activating channelrhodopsin (ChR2). We injected a Cre-dependent adeno-associated virus (AAV) encoding the activating ChR2 into the dmPFC of *PV-Cre* mice and implanted a wireless LED cannula over AAV injection sites in the dmPFC (Supplementary Fig. 3). Adult mice attached to a wireless battery and receiver were tested in a variety of social behavior and control tests, including open-field, reciprocal interactions, and the 3-chamber test (Fig. 2a). We validated that 40 Hz blue light stimulation can reliably induce spiking in dmPFC PVIs (Fig. 2b, c).

In a reciprocal interactions test, 3 s of 40 Hz blue light was randomly triggered with 8 to 15 s between pulses during a 5-min testing session. Mice were tested under both "on" and "off"

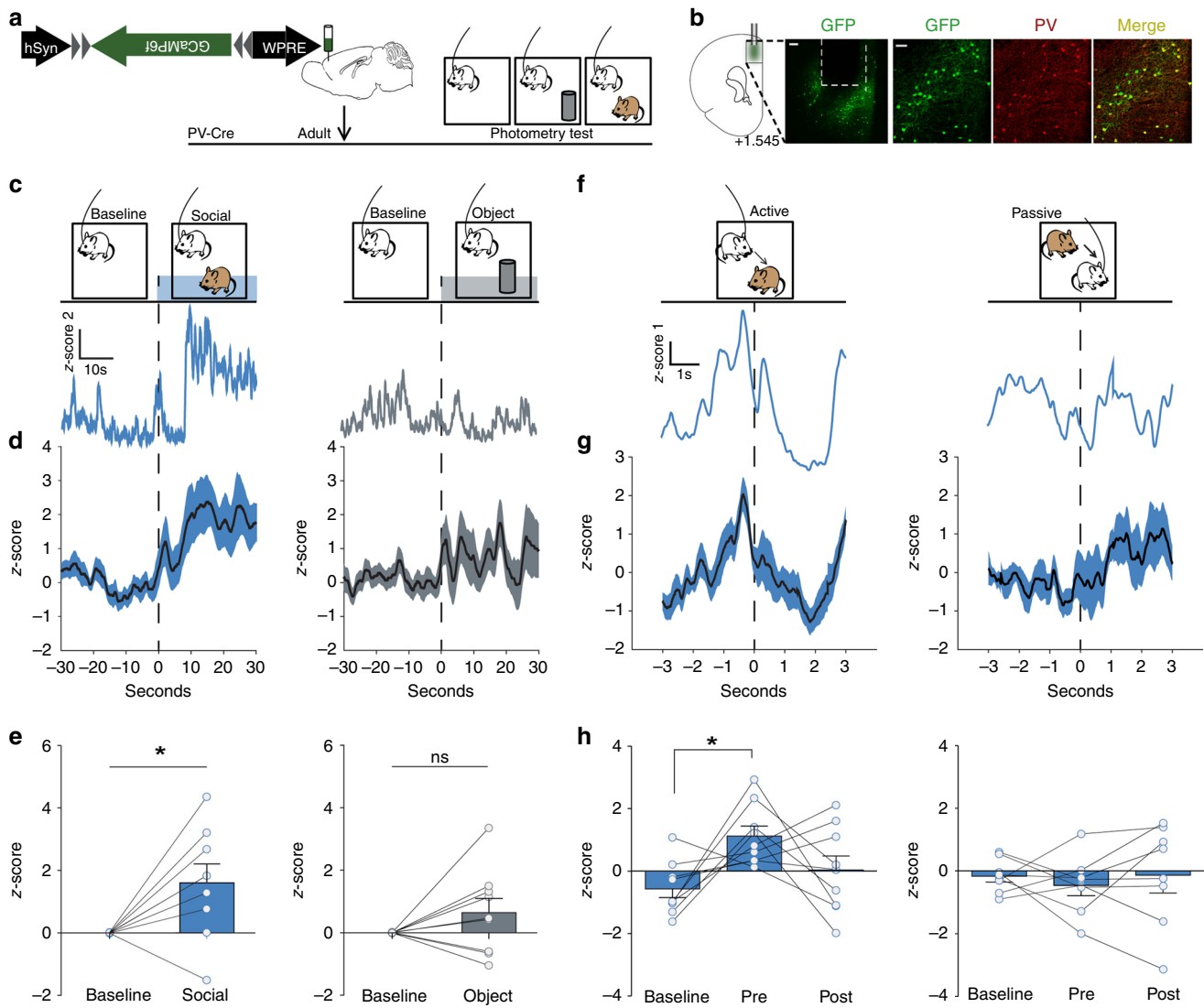

**Fig. 1 Short activation of dmPFC PVIs precedes an active social approach. a** Timeline showing injection of GCaMP6f in adult dmPFC in *PV-Cre* mice and subsequent behavioral testing paradigm for fiber photometry imaging (order of object and social exploration was counterbalanced). **b** Example location of fiber ferrule and GCaMP6f expression in mouse PFC (left) and co-localization of GCaMP6f expression (green) and PV staining (red). (left, scale bar = 100 μm; right, scale bar = 50 μm). **c**, **d** GCaMP6f signals of dmPFC-PVIs show increased mean *z*-score 30 s after introduction of a novel mouse (left), but not a novel object (right). **c** Example GCaMP signal traces. **d** Mean +/− SEM smoothed signal. **e** Mean dmPFC-PVI GCaMP6f signal for each mouse comparing baseline and post stimulus introduction for social and object (Wilcoxon signed-rank test, $n = 9$ mice, social: *$p = 0.03$, object: $p = 0.25$). **f** Top: active behavior, defined as behavior initiated by the focal mouse (left) and passive behavior, defined as behavior initiated by the stimulus mouse (right) were scored during photometry imaging. Bottom: Representative responses from the first active and first passive bout shows an increase in dmPFC-PVI activity immediately prior to the active bout initiation. No change is seen in the passive bout. **g** Mean +/− SEM smoothed signal for the first active and passive bout. **h** Mean dmPFC-PVI GCaMP6f signal for each mouse comparing baseline (−3 to −2 s before active/passive initiation) pre (−1 to 0 s) and post (0–1 s). (One-way repeated measures ANOVA, active (left): $F_{(2,8)} = 4.67$, *$p = 0.02$, Tukey post hoc tests, active baseline vs pre, *$p < 0.05$, $n = 9$ mice. Passive (right): $F_{(2,7)} = 0.27$, $p = 0.77$, $n = 8$ mice. All error bars reflect +/− SEM. See related Supplementary Figs. 1 and 2. Source data is available as a Source Data file.

conditions, counterbalanced, and behavior was scored for 8 s following pulse initiation. In the reciprocal interactions test, mice showed increased duration of active behaviors during the "on" condition, but there were no differences in the duration of passive or orienting behaviors (Fig. 2d, left: linear mixed model, effect of treatment (light on vs. off), $p = 0.04$, planned post hoc tests: active on vs off, $p = 0.01$, orient on vs. off $p = 0.40$, passive on vs. off $p = 0.30$). The specific behaviors that most commonly followed the initial orienting response to the stimulus (Supplementary Fig. 4a, b) were "approach" and "nose-to-nose" (active behaviors). These were also the behaviors that increased between on and off conditions (Fig. 2d, right: paired *t* tests, nose-to-nose

$p = 0.01$, approach $p = 0.05$). There was no overall effect on velocity in an open-field test, using the same 3-s-stimulation and 8-s-measuring protocol (Fig. 2e, paired *t* test, $p = 0.10$).

To further test the relationship between short dmPFC-PVI activity and subsequent active social approach in a more controlled setting, we tested mice in a 3-chamber test for sociability, in which a mouse chooses between a social target and an object, and time spent investigating both is measured and compared[24]. Three seconds of 40 Hz light pulses were triggered when mice entered the center chamber, behavior in the following 8 s was categorized into either object chamber entry, social chamber entry, or remain in the center chamber. Mice were tested

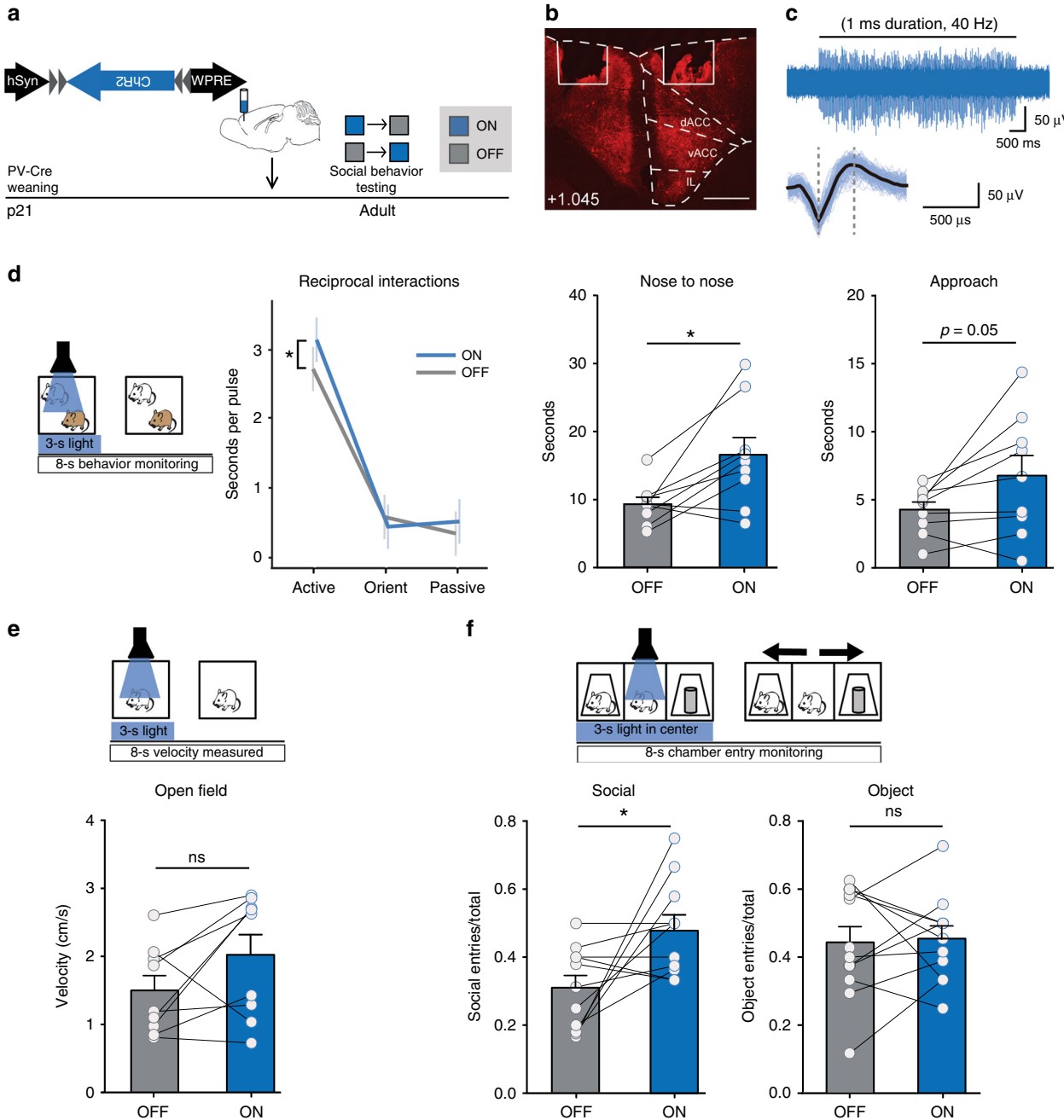

**Fig. 2 Short optogenetic activation of dmPFC PVIs promotes subsequent social approach. a** Timeline showing injections of Cre-dependent Channelrhodopsin (ChR2) in *PV-Cre* mice and subsequent social behavior in adult mice. Light order was counterbalanced. **b** Representative viral transduction (ChR2-mCherry) and LED location. Scale bar = 500 μm. **c** Representative in vivo spike activity of ChR2-expressing dmPFC-PVIs upon 40 Hz stimulation (top), and averaged spike following 40 Hz stimulation showing characteristic narrow spike shape (bottom). **d** Three seconds of light were randomly triggered at least 8 s apart over the course of a 5 min reciprocal interaction trial with an unfamiliar age-, sex-, and strain-matched mouse. Behavior was measured for 8 s following initiation of the light pulse (left). Duration of active behavior per pulse was increased in 'on' compared with 'off' conditions, while orienting to the stimulus and passive behavior were not changed (middle, linear mixed model effect of light, $p = 0.04$, planned post hoc tests: active on vs off, *$p = 0.01$, orient on vs. off $p = 0.40$, passive on vs. off $p = 0.30$, $n = 9$ mice). By animal statistics reveal a significant increase in duration of nose-to-nose investigation and Approach (right, paired $t$ tests, nose-to-nose: $t = 3.14$, df = 8, *$p = 0.01$, Approach: $t = 2.25$, df = 8, $p = 0.05$, $n = 9$ mice). **e** Using the same stimulation protocol as in (**d**), velocity was measured in the 8 s following light-stimulation initiation. There were no changes in velocity (paired $t$ test, $t = 1.88$, df = 8, $p = 0.10$, $n = 9$ mice). **f** Three seconds of light stimulation were triggered when the mouse was in the center chamber, and subsequent entries into either the social chamber, the object chamber, or neither were measured in the 8 s following light initiation (top) in a 3-chamber. Bottom: social entries (left) but not object entries (right) increase following 3 s 40 Hz light stimulation in the 3-chamber test (paired $t$ test, Social: $t = 2.54$, df = 10, *$p = 0.03$ Object: $t = 0.14$, df = 9, $p = 0.89$, $n = 11$ mice). Error bars reflect +/− SEM. See related Supplementary Figs. 3 and 4. Source data are available as a Source Data file.

under both "on" and "off" conditions. While mice showed a social preference in both on and off conditions with no significant difference (Supplementary Fig. 4c, paired $t$ test, $p = 0.95$), mice made significantly more entries into the social chamber, during the "on" phase (Fig. 2f, paired $t$ test, social, $p = 0.03$, $n = 11$). Object chamber entries were not significantly changed (Fig. 2f, paired $t$ test, object $p = 0.89$, $n = 11$). Mice injected with eYFP virus did not show an effect of light in entries to the social or object chamber (Supplementary Fig. 4d–f, paired $t$ test, social, $p = 0.78$, object, $p = 0.3$, $n = 9$). Collectively, these studies suggest that brief dmPFC-PVI activation, which is associated with active bout initiation during a naturalistic social encounter, is sufficient to promote social approach.

**Suppression of dmPFC PVIs causes social deficits**. Since dmPFC-PVIs are active during social exposure and prior to an active bout (Fig. 1), and brief dmPFC-PVI activation promotes social approach (Fig. 2), we next set out to test the causal relationship between physiologic levels of dmPFC-PVI activity during a social task and active social behavior in the 3-chamber test. To this end, we employed a chemogenetic approach to selectively suppress the activity of dmPFC PVIs using an inhibitory (i) DREADD (designer receptors exclusively activated by designer drugs)[25,26]. Because of recent evidence that the ligand for DREADDS, CNO (clozapine-$N$-oxide), could have endogenous effects in mouse brain[27], we confirmed that intraperitoneal injection of CNO (10 mg/kg) in the absence of the DREADD virus had no effect on social behavior in a 3-chamber test (Supplementary Fig. 5a, b, two-way repeated measures ANOVA effect of drug, time in social chamber $p = 0.42$, social interaction per chamber entry, $p = 0.47$), anxiety or locomotion in an open-field test (Supplementary Fig. 5c, two-way repeated measures ANOVA, distance traveled: effect of the drug, $p = 0.74$, time in the center: effect of the drug, $p = 0.24$), or behavior in the light–dark box (Supplementary Fig. 5d, paired $t$ test, time in the light: $p = 0.81$, time in the dark: $p = 0.88$). Our viral injections in $PV$-$Cre$ mice demonstrated highly specific expression of iDREADD in PVIs (Fig. 3b. $82.54 \pm 7.4\%$ of iDREADD-positive cells co-expressing PV), and successfully target dmPFC (Fig. 3b; Supplementary Fig. 6). We also validated that bath application of CNO significantly decreased the membrane potential of dmPFC-PVIs (Fig. 3c, paired $t$ test $p < 0.001$), confirming reliable suppression by the hM4Di-selective agonist, CNO.

We then set out to test whether activity of dmPFC-PVIs is necessary for appropriate social behavior using a repeated measures behavioral design to assess the effect of acute dmPFC-PVI suppression in a battery of behavior tests, including the 3-chamber test for sociability[24], an open field to test activity and anxiety behavior, and a light–dark box test to assess anxiety behavior (Fig. 3a). We injected CNO or saline (SAL) intraperitoneally 30 min prior to testing. Acute dmPFC-PVI suppression led to a trending decrease in time in the social chamber (Fig. 3d (top), two-way repeated measures ANOVA effect of drug, $p = 0.07$) and a significant decrease in time spent interacting for every social chamber entry (Fig. 3d (bottom), two-way repeated measures ANOVA, effect of the drug $p = 0.005$), suggesting a decrease in active social behavior. However, iDREADD mice still retain social preference under CNO (paired $t$ test, $p = 0.004$). There were no changes in control behaviors used to assess activity and anxiety (Fig. 3e, open-field two-way repeated measures ANOVA, effect of the drug, distance traveled $p = 0.92$, time in the center $p = 0.66$, Fig. 3f, light–dark box: paired $t$ test, time in the light: $p = 0.31$, time in the dark: $p = 0.28$). Collectively, these findings demonstrate that PVI suppression in dmPFC disrupts social behavior, without affecting exploration, activity, or anxiety,

suggesting that proper dmPFC-PVI activity is physiologically necessary for normal social behavior.

**JSI disrupts dmPFC-PVI activity during active social behavior**. Since dmPFC-PVIs respond to social signals in adulthood (Fig. 1), promote active social investigation in adults (Fig. 2) and are necessary for normal social behavior (Fig. 3), we hypothesized that dmPFC-PVIs may require social experience during sensitive developmental windows to establish their adult patterns of activity and to regulate adult social behaviors. We leveraged a previously established paradigm showing that social isolation during a 2-week juvenile window (p21–35) leads to decreased social investigation in adult male mice[3]. We confirmed that 2 weeks of juvenile social isolation in male mice (jSI) leads to long-lasting social deficits (Supplementary Fig. 7a, b $t$ test, $p = 0.02$). Importantly, this type of manipulation did not lead to aggression upon re-introduction or wounding within the cage. There were trending increases of serum corticosterone levels at p35 (Supplementary Fig. 8a, $t$ test $p = 0.07$); however, this was not maintained into adulthood (Supplementary Fig. 8b, $t$ test $p = 0.78$), showing no persistent alterations in the hypothalamic–pituitary–adrenal axis in adult jSI mice. We chose to focus our study on male mice because jSI has not been shown to induce long-lasting social deficits in female rodents[28], despite effects across other behavioral domains[29]. However, since this manipulation involves re-grouping at p35 and subsequent co-housing into adulthood, we controlled for any potential effects of unfamiliar cagemates independent of social isolation by adding a "shuffled" group of mice that were group housed from p21–35 and re-grouped with unfamiliar age-, strain-, and sex-matched individuals at p35, then tested in adulthood. We found that shuffled mice did not show persistent social deficits (Supplementary Fig. 7a, c, $t$ test, $p = 0.36$). We confirmed that p21–35 is indeed a sensitive window for social experience by isolating adult mice for 2 weeks, re-housing them for 1 month, and then testing social behavior. Adult isolated mice do not show persistent social deficits (Supplementary Fig. 7a, d, $t$ test $p = 0.74$). These findings confirm a sensitive window for social experience-dependent adult social behavior development.

To test whether dmPFC-PVIs are impacted by jSI, we used fiber photometry to assess dmPFC-PVI activity in adult jSI mice during a social or object exploration task (Supplementary Fig. 9; Fig. 4a). Overall response to the presence of an unfamiliar age-, sex-, and strain-matched mouse in jSI mice showed a similar pattern to group housed (GH) mice (Supplementary Fig. 10, $t$ test social: $p = 0.005$, object: $p = 0.66$, $n = 8$). However, behaviorally, joint probability distributions of transitions showed jSI mice exhibit distinct transition frequencies between subclasses of behavior. Visual inspection showed a distinct set of commonly connected behaviors. While no two specific behaviors showed a significant difference in transition frequency between groups, we observed a decrease in transition frequencies between active behaviors in jSI mice, and an increase in transitions between active and passive behaviors compared with GH mice (Fig. 4b; Supplementary Fig. 11). There was an overall decrease in duration of active social behavior in the reciprocal interactions task between jSI and GH mice (Fig. 4c, two-way mixed model ANOVA, Bonferroni post hoc test, active behavior duration $p < 0.05$). Given this behavioral disparity between jSI and GH mice, we examined whether jSI mice showed a similar dmPFC-PVI activity pattern surrounding initiation of active and passive social bouts as was observed in GH mice. JSI mice did not show a pre-active bout initiation increase in dmPFC-PVIs prior to the first active bout, unlike GH mice (Fig. 4d, e, left: baseline vs. pre, Tukey post hoc test, $p > 0.05$). However, dmPFC-PVIs did show a response to social signals briefly after mice are passively explored (Fig. 4d, e, right: first

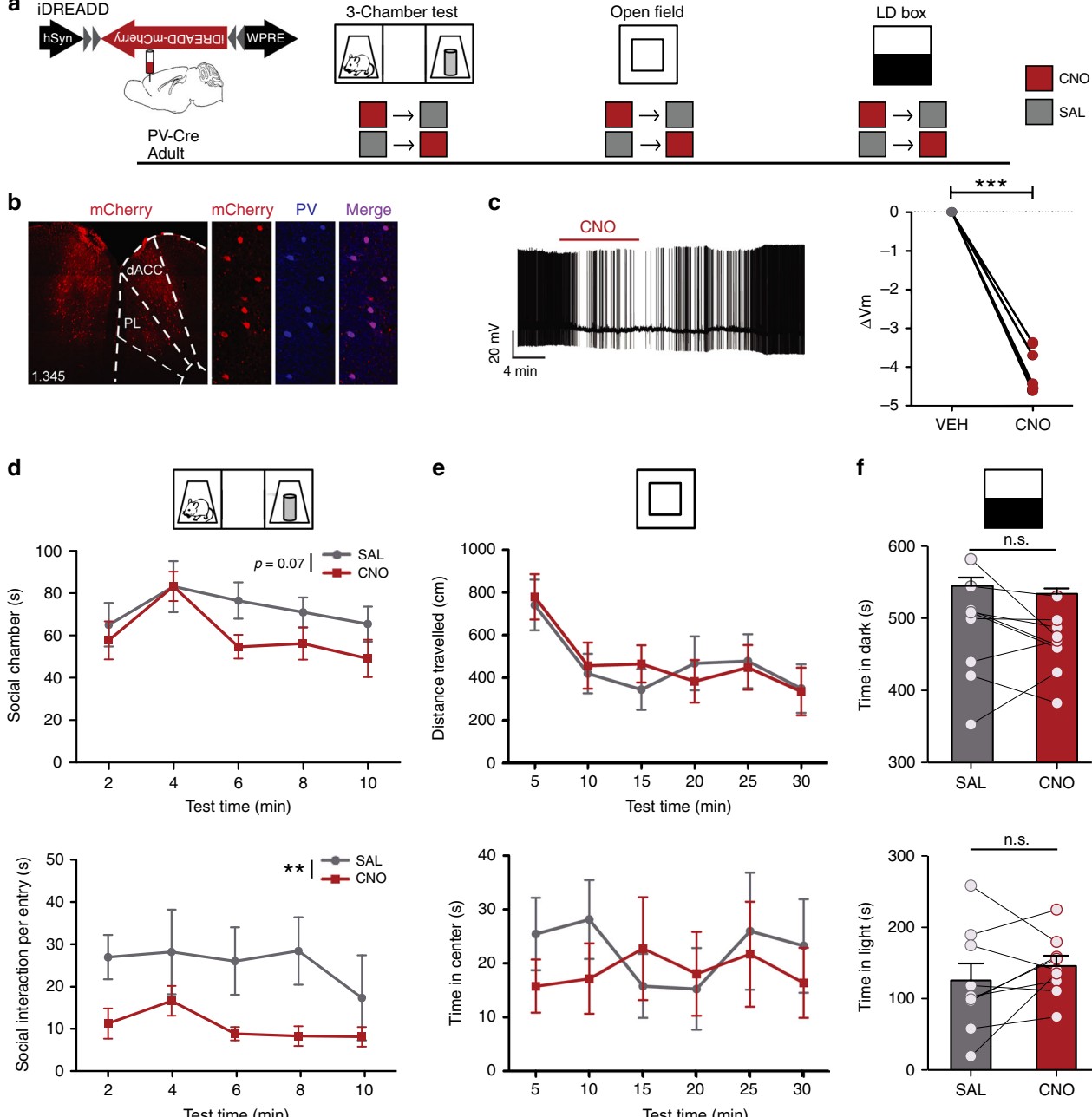

**Fig. 3 Chemogenetic suppression of adult dmPFC-PVIs activity disrupts social behavior. a** Timeline showing injection of Cre-dependent inhibitory DREADD (iDREADD) virus to dmPFC of adult *PV-Cre* mice and subsequent repeated measures behavioral design. For CNO (red square) and SAL (gray square) injections, each mouse is tested under both CNO and SAL conditions, and the order is counterbalanced for each behavior test. **b** Representative injection of inhibitory DREADD injected in dmPFC of *PV-Cre* mouse (left, scale bar = 100 μm) and co-localization of DREADD virus expression (mCherry, red) with PV staining (blue) (right, scale bar = 50 μm). **c** Validation of iDREADD in dmPFC-PVI. Left: representative trace of whole-cell recording from a PVI in adult dmPFC slice upon CNO bath application. Right: change in membrane potential from vehicle after CNO application. Repeated measures *t* test, *t* = 16.68, df = 5, ***p < 0.001, n = 6 cells from three mice. **d** Three-chamber tests show trending decreases in time in social chamber (top), and significant decreases in time interacting per social chamber entry (bottom) when PVIs are suppressed with CNO, compared with saline. (n = 9 mice. Two-way repeated measures ANOVA, effect of the drug: F(1,8) = 4.39, p = 0.07 (top), F(1,8) = 14.28, p = 0.005 (bottom)). **e** No effect of the drug was seen on distance traveled (top) or time in the center vs. periphery (bottom) in an open-field test (two-way repeated measures ANOVA, distance traveled: effect of the drug, F(1,7) = 0.01, p = 0.92, time in the center: effect of the drug, F(1,7) = 0.21, p = 0.66, n = 8 mice), or in **f** the light–dark (LD) box test in either time in the dark (top) or time in the light (bottom), (paired *t* test, time in light: *t* = 1.089, df = 8, p = 0.31, time in the dark: *t* = 1.16, df = 8, p = 0.28, n = 9 mice). All error bars reflect + /− SEM. See related Supplementary Figs. 5 and 6. Source data are available as a Source Data file.

passive bout: pre vs. post, Tukey post hoc test, *p* < 0.001). GH, but not jSI mice, showed a significant change from baseline during the first active bout, while jSI mice showed a significant change from baseline only after the first passive bout (Fig. 4f). This is specific to

the first active and passive bouts (Supplementary Fig. 12). JSI mice did not show significant changes in dmPFC-PVI activity in active object exploration (Supplementary Fig. 13). We conducted an additional experiment comparing dmPFC-PVI activity prior to a

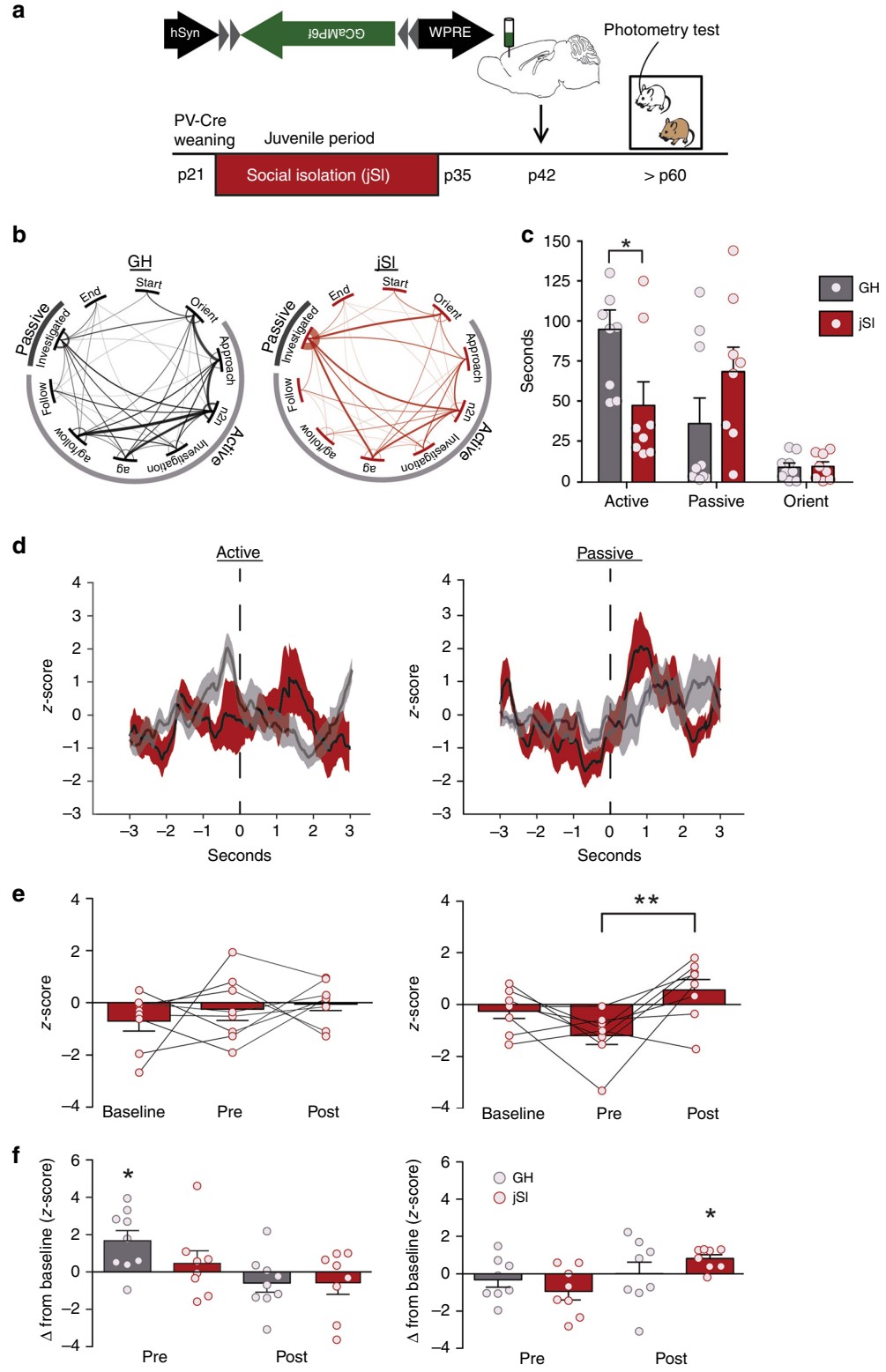

social chamber entry in a separate cohort of animals tested in the 3-chamber test to achieve more precise timing of social behavior initiation and found that prior to all social chamber entries, GH mice show increased dmPFC-PVI activity compared with jSI mice (Supplementary Fig. 14). Collectively, these findings demonstrate that dmPFC-PVIs show distinct activity patterns during a reciprocal interactions test and a 3-chamber test in jSI mice, and are not active prior to the first active social bout or prior to social

chamber entries, implicating a crucial role of juvenile social experience in coupling dmPFC-PVI activity with active social approach behavior.

**JSI alters development of dmPFC-PVI excitability and drive**. Altered in vivo calcium signals during social behavior may reflect underlying alterations in the electrophysiological properties of dmPFC-PVIs and altered synaptic inputs. Adult differences seen

**Fig. 4 Juvenile social isolation alters adult dmPFC-PVI activity during reciprocal social interaction. a** Timeline showing weaning at p21, and subsequent 2 weeks of juvenile social isolation (jSI), followed by injection of GCaMP6f in the dmPFC of *PV-Cre* jSI mice at p42, and subsequent reciprocal interaction behavioral testing paradigm for fiber photometry imaging. **b** Transition matrices between specific behaviors reveal significant differences between the joint-distribution matrix between jSI and GH mice ($p < 0.05$), suggesting distinct sequences of behaviors between two groups. **c** Behavior categories during photometry imaging show decreased active exploration in jSI vs. GH mice (two-way mixed model ANOVA, effect of housing, $F_{(1,15)} = 1.42$, $p = 0.25$, housing x behavior interaction, $F_{(2,15)} = 9.57$, $p = 0.03$, Bonferroni post hoc tests, active behavior *$p < 0.05$, passive, Orient, $p > 0.05$, $n = 8$ mice jSI, $n = 9$ mice GH). **d** Mean and SEM smoothed signal for the first active and passive bout in jSI mice (red) and GH (gray: from Fig. 1). **e** jSI mice do not show increased pre-active GCamp6f signal in dmPFC-PVIs (left) (baseline vs. pre, Tukey post hoc test $p > 0.05$), however, dmPFC-PVIs do respond during the first passive encounter, with a significant increase in activity (pre vs. post, Tukey post hoc test, **$p < 0.001$). **f** Change in z-score from baseline in the first active encounter shows a significant difference from zero only in GH mice during the pre-active time bin. Change in z-score from baseline in the first passive encounter shows a significant difference from zero only in the post-passive time bin. (Wilcoxon signed-rank test, active (left): GH pre, *$p = 0.03$, passive (right): *$p = 0.02$, $n = 9$ GH mice, $n = 7$ jSI mice). All error bars reflect $+/-$ SEM. See related Supplementary Figs. 7–14. Source data are available as a Source Data file.

in dmPFC-PVI functioning may emerge during the isolation period due to lack of social-dependent activation of dmPFC-PVIs, or may emerge later, reflecting a deviation from a normal developmental trajectory. To test this hypothesis, we assessed the effect of jSI on dmPFC-PVIs by whole-cell patch-clamp recordings from fluorescently labeled PVIs at p35 and in adults (Fig. 5a). We compared intrinsic excitability and synaptic properties in jSI and GH mice at both time points.

Intrinsic excitability of dmPFC-PVIs measured in the presence of synaptic blockers showed a strong increase in spike frequency, especially at higher current steps, between p35 and adulthood in GH mice (Fig. 5b, left: two-way mixed model ANOVA, age factor: $p = 0.0035$, interaction with current step $p < 0.0001$). This suggests that even between adolescence and adulthood, dmPFC-PVIs in GH mice undergo substantial maturation. Strikingly, in jSI mice this developmental change was completely absent: input–output curves were not changed between p35 and p60 (Fig. 5b, middle: two-way mixed model ANOVA, age factor: $p = 0.15$, interaction with current step $p = 0.80$). At p35, there were no differences between jSI and GH, however, in adulthood, GH mice showed a significantly higher spike frequency than jSI mice (Fig. 5b, right: two-way ANOVA, housing factor: $p = 0.03$, age factor: $p = 0.0004$, Bonferroni post hoc tests at p60 $p < 0.05$). This finding suggests dmPFC-PVIs in jSI show a developmental "freeze" pattern, failing to develop normal excitability even after the end of the isolation. We also examined spontaneous excitatory postsynaptic currents (sEPSCs) and spontaneous inhibitory postsynaptic currents PSCs (sIPSCs). sEPSCs and sIPSCs showed a developmental decrease in frequency as well as a significant main effect of housing, showing a decrease in sEPSC frequency between jSI and GH (Fig. 5c, left: two-way ANOVA, age factor: $p < 0.001$, housing factor: $p = 0.01$). There was also a significant decrease in sIPSC frequency across ages and a trending interaction between age and housing, indicating slightly lower initial sIPSC frequency in jSI mice at p35, but moderately increased sIPSC frequency by adulthood (Fig. 5c, middle: two-way ANOVA, age factor: $p < 0.001$, housing by age interaction: $p = 0.07$). There was no significant effect of housing or age on the sEPSC amplitude (Supplementary Fig. 15c, left: two-way ANOVA, age factor: $p = 0.22$, housing factor: $p = 0.26$, interaction: 0.74). There was a significant decrease in sIPSC amplitude across development, but no effect of housing (Supplementary Fig. 15c, right: two-way ANOVA, age factor: $p = 0.0012$, housing factor: $p = 0.65$, interaction: $p = 0.37$). The ratio of sEPSC/sIPSC frequency, an indicator of overall "drive" received by dmPFC-PVIs, showed an increase from adolescence to adulthood in GH mice, and no developmental change in jSI mice (Fig. 5c, right: two-way ANOVA, housing by age interaction: $p = 0.0005$). This pattern mimics the developmental "freeze" phenotype, indicating a late developmental increase in excitatory/inhibitory

drive occurring in GH mice but not in jSI mice leading to a significant difference in this ratio by p60 (Bonferroni post hoc test, $p < 0.0001$). Miniature excitatory postsynaptic currents (mEPSCs) and miniature inhibitory postsynaptic currents (mIPSCs) also showed developmental decreases between p35 and p60 independent of housing (Supplementary Fig. 15b, two-way ANOVA, age factory: mEPSC, $p < 0.001$, mIPSC, $p < 0.001$). Our findings strongly suggest that jSI results in action potential-dependent, network-level alterations in the adult animal. Furthermore, the observed decreases in intrinsic excitability and ratio of excitatory to inhibitory input drives onto adult dmPFC-PVIs in jSI mice compared with GH mice suggest a decreased activation pattern of dmPFC-PVIs in adult jSI mice, suggesting social experience during this window is required for dmPFC-PVIs to develop normal adult activity patterns.

**Activating dmPFC-PVIs rescues social deficits induced by jSI.** Finally, we tested whether increasing activity of dmPFC-PVIs through excitatory DREADDs (eDREADDs) can rescue social behavior deficits in adult mice, induced by jSI. We first validated the dmPFC-PVI specificity of the eDREADD virus (Fig. 6a), confirmed targeting of our injection site to the dmPFC (Fig. 6b; Supplementary Fig. 16), and validated, by quantification of dmPFC-PVI expression of the activity-regulated early response gene, egr-1, that a 1 mg/kg intraperitoneal injection of CNO can reliably activate dmPFC-PVIs expressing the eDREADD virus (Supplementary Fig. 17a–c, $t$ test, $p < 0.0001$). We then tested whether chemogenetic activation of dmPFC-PVIs in adult mice could ameliorate jSI-induced social deficits. *PV-Cre* male GH/jSI mice were bilaterally injected with Cre-dependent eDREADD virus at p42. At p65, we assessed the effect of acutely exciting dmPFC-PVIs on social behavior by injecting CNO or saline (SAL) intraperitoneally 30 min before the 3-chamber test in a counter-balanced design. We found that jSI mice that were treated with CNO showed significantly higher amounts of social investigation compared with SAL treated mice in the first 2 min of the 3-chamber test, with no change in the time spent investigating an object, indicating an increase in sociability (Fig. 6c, d, paired $t$ test, social: $p = 0.02$, object: $p = 0.3$). JSI mice treated with saline did not show a significant social preference, and this deficit was rescued under CNO treatment (Fig. 6c, two-way repeated measures ANOVA, drug x stimulus interaction $p = 0.03$, Bonferroni post hoc tests: SAL: $p > 0.05$, CNO: $p < 0.001$). GH mice showed a significant social preference when treated with either SAL or CNO, indicating no effect of activating dmPFC-PVIs in GH animals (Fig. 6c, two-way repeated measures ANOVA, drug x stimulus interaction $p = 0.73$, Bonferroni post hoc tests: SAL: $p < 0.05$, CNO: $p < 0.01$). JSI mice treated with CNO showed increased time in the social chamber and increased social interaction per

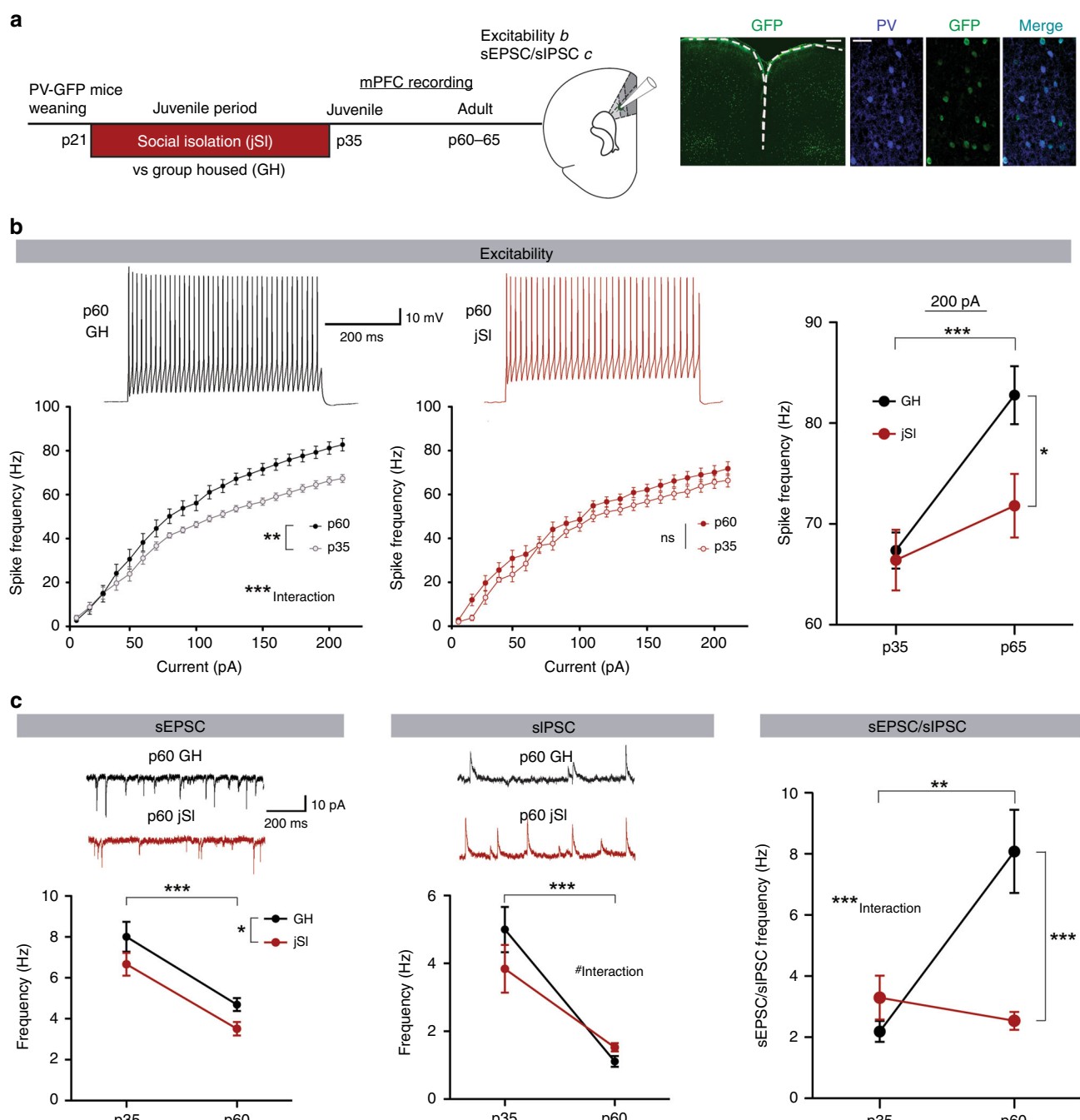

**Fig. 5 Juvenile social isolation causes altered maturation of dmPFC PVIs, leading to long-lasting reduction of excitability and input drive of adult dmPFC PVIs. a** Timeline of juvenile social isolation (jSI) or group housing (GH) followed by whole-cell patch clamp from dmPFC PVIs at p35 or adulthood (left). Right: GFP+ cells (green) co-localize with PV+ cells (blue). Scale bar: left: 300 μm right 50 μm. **b** Top: Intrinsic excitability of dmPFC PVIs. Representative traces, 200 pA.. Bottom left: adult GH mice showed increased spike frequency at higher current steps (two-way mixed ANOVA, Age factor, $F_{(1,40)} = 9.36$, **$p = 0.0035$, age x current $F_{(20,40)} = 5.91$, ***$p < 0.0001$). (bottom, middle) In jSI mice, there were no developmental differences (two-way mixed ANOVA, age factor $F_{(1,37)} = 2.15$, df = 1, $p = 0.15$, age x current $F_{(20,37)} = 0.73$, df = 20, $p = 0.7977$). Right: At 200 pA, spike frequency was significantly lower only in adult jSI mice (two-way ANOVA, housing factor, $F_{(1,77)} = 4.53$, *$p = 0.04$, age factor, $F_{(1,77)} = 13.75$, ***$p = 0.0004$, Bonferroni post hoc tests p60: $p < 0.05$, p35: $p > 0.05$). $n = 19$ cells, four mice (p36, GH/jSI), 20 cells, five mice (p60, jSI) and 23 cell five mice (p60, GH). **c** Upper: representative postsynaptic current (PSC) traces. Lower left: sEPSC frequency significantly decreased across development and was lower in jSI (two-way ANOVA, age factor, $F_{(1,81)} = 43.75$, ***$p < 0.001$, housing factor, $F_{(1,81)} = 6.68$, *$p = 0.01$, p35 sEPSCs: $n = 20$ cells, five mice (GH), $n = 17$ cells, five mice (jSI), adult sEPSCs: $n = 26$ cells, seven mice (GH), 22 cells, six mice (jSI)). Lower middle: sIPSC frequency also decreased in adults; however, there was a trending interaction (two-way ANOVA, age factor, $F_{(1,81)} = 49.59$, ***$p < 0.0001$, age x housing $F_{(1,81)} = 3.20$, #$p = 0.07$, p35 sIPSCs: $n = 20$ cells, five mice (GH), $n = 17$ cells, five mice (jSI), adult sIPSCs: $n = 26$ cells, seven mice (GH), $n = 22$ cells, six mice (jSI)). Right: the developmental change of the sEPSC/sIPSC frequency ratio showed a strong interaction (two-way ANOVA, age x housing, $F_{(1,81)} = 13.06$, ***$p = 0.0005$), indicating an absent developmental increase in jSI mice (Bonferroni post hoc test p60, ***$p < 0.0001$). Error bars reflect $+/-$ SEM. Supplementary Fig. 15. Source data are available as a Source Data file.

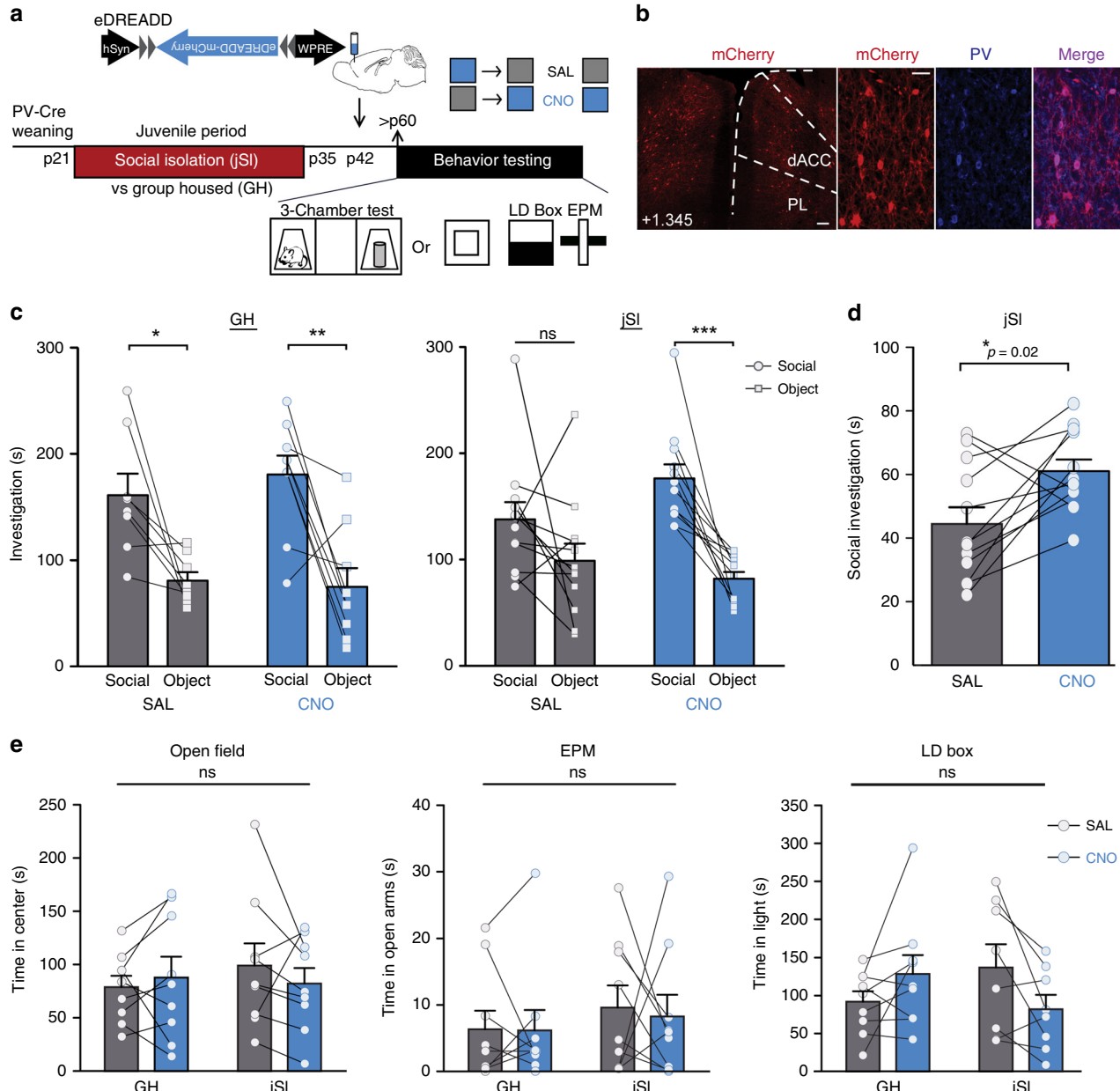

**Fig. 6 Chemogenetic activation of adult dmPFC-PVIs mitigates social deficits induced by juvenile isolation. a** Timeline showing juvenile social isolation (jSI), Cre-dependent excitatory DREADD (eDREADD) viral injection (p42) in *PV-Cre* mice, and sociability and anxiety behavior testing (adult, p60–80). **b** Immunohistochemistry validation of co-localization between mCherry (eDREADD, red) and parvalbumin (PV) antibody (blue). Scale bar left = 100 μm. Scale bar right = 50 μm. **c** Full behavioral results from the 3-chamber test for sociability show that jSI mice treated with SAL do not show a social preference; however, mice treated with CNO show a significant social preference. GH mice show a significant social preference under both SAL and CNO conditions (two-way repeated measures ANOVA, GH: drug x stimulus interaction F(1,7) = 0.13, p = 0.73 jSI: drug x stimulus interaction, F(1,11) = 6.28, *p = 0.03 followed by post hoc Bonferroni corrected t tests, *p < 0.05, **p < 0.01, ***p < 0.001). n = 12 mice jSI, 8 mice GH. **d** Juvenile socially isolated JSI adult mice show a significant increase in social interaction in the first 2 min of behavior testing when treated with CNO compared with SAL (paired t test, t = 2.75, df = 11, *p = 0.02). **e** Neither jSI nor dmPFC-PVI activation with CNO has significant effects on anxiety behavior in the open field (mixed effects ANOVA, OF: effect of housing, p = 0.73, F(1,16) = 0.12, effect of the drug, F(1,16) = 0.12, p = 0.74, interaction, F(1,16) = 1,26, p = 0.28), elevated-plus maze (EPM: effect of housing, F(1,19) = 0.68, p = 0.42, effect of the drug, F(1,19) = 0.06, p = 0.81, interaction, F(1,16) = 0.04, p = 0.85), or the light–dark box (LD, effect of housing, F(1,16) = 0.09, p = 0.77, effect of the drug, F(1,16) = 1.03, df = 1, p = 0.32, interaction F(1,16) = 3.36, p = 0.09). All error bars reflect +/− SEM. See related Supplementary Figs. 16 and 17. Source data are available as a Source Data file.

social chamber entry across the length of the test (Supplementary Fig. 17d, left: two-way repeated measures ANOVA, social chamber effect of drug p = 0.04, social interaction per entry effect of the drug p = 0.006) while group housed mice showed no change (Supplementary Fig. 17d, right: two-way repeated measures ANOVA, social chamber effect of the drug p = 0.78, social

interaction per entry effect of the drug p = 0.83). There were no significant changes in anxiety, measured in the open field, the elevated-plus maze, and the light–dark box. We conclude that no strong anxiety phenotypes are induced by jSI, and there is no effect of dmPFC-PVI excitation on anxiety behavior (Fig. 6e). Our experiments demonstrate that increasing adult dmPFC-PVI

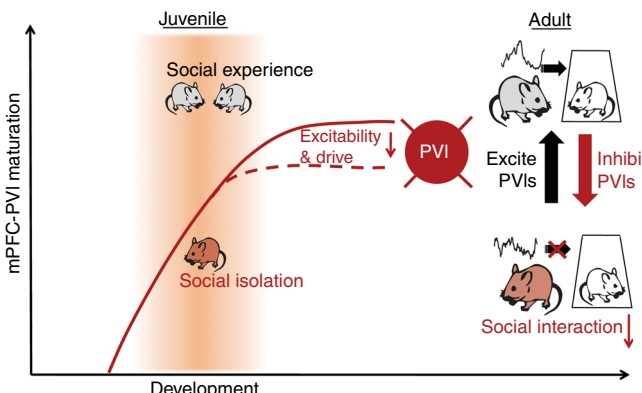

**Fig. 7 Summary Scheme: PFC-PVIs show decreased excitability and drive following juvenile social isolation, a manipulation that leads to decreased social interaction.** In group housed adults, short dmPFC-PVI activation induced by optogenetics mirrors naturally observed pre-active dmPFC-PVI activity, and leads to social approach. This relationship is not observed in juvenile socially isolated mice. Decreased social interaction can be induced in group housed animals by inhibiting PFC-PVI in adult mice; social interaction of isolated mice can be mitigated by increasing PFC-PVI activity.

activity through eDREADDs ameliorates jSI-induced social behavior deficits.

## Discussion

Demonstrating a cellular mechanism linking experience during sensitive windows with adult outcomes has been a major focus of sensory encoding research; however, translating these foundational ideas to study the development of complex behaviors relevant to psychiatric research has been challenging. Here, we demonstrate that juvenile social experience is required for PVIs in mouse dmPFC to develop typical adult activity patterns, in which brief activity triggers subsequent active social approach (Fig. 7). Our study focused on juvenile social experience because we showed that adult social isolation for 2 weeks does not impact social behavior, which has also been observed by others[30]. While social behavior deficits can be induced in adult mice if subjected to extensive social isolation (8–10 weeks), this effect is reversible by 4 weeks of re-grouping in contrast to the persistent jSI effect[31]. Many previous studies focusing on social isolation wean rodents into isolation and test in adulthood without re-housing in a social setting[29,32]. While these foundational studies have instructed our understanding of the importance of social experience, they do not shed light on specific windows of social deprivation and cannot tease apart which phenotypes are persistent and which are ameliorated with social housing. Early adolescence is a unique developmental window, during which peer social interactions are particularly salient across mammals. In rodents, the post-weaning juvenile period is characterized by specific sets of social interactions such as play behavior[33,34]. Our findings uncover important insights linking social experience during sensitive windows with dmPFC local circuit maturation in control of social behavior.

While many studies group "social interaction" as one cohesive unit, social interactions are characterized by a diverse range of behaviors. Our study demonstrated that dmPFC-PVIs are selectively activated prior to an active, but not a passive bout (Fig. 1). This finding adds to previous experiments, which have examined dmPFC-PVI activity, and seen increases occurring during a social encounter[21,35], or during an approach and encounter[36] but have not examined the window prior to the initiation of the bout. This highlights the importance of studying activity preceding initiation

of behavior, which may shed light on circuits that can regulate downstream mechanisms to influence behavior. This pre "active" increase in dmPFC-PVI activity may be important for gating specific projection neurons while inhibiting specific "anti-social" subcortical projections from dmPFC[37–39], leading to selective activation of circuits necessary for subsequent social interaction. Alternatively, given that brief optogenetic modulation of dmPFC-PVI at gamma frequency triggers active social approach, rhythmic dmPFC activity in the gamma oscillation range may be necessary for expression of normal social approach, as it is for attentional behavior, without impacting the overall excitation of dmPFC region (Supplementary Fig. 17c)[40].

Our study demonstrated that juvenile socially isolated mice exhibit a decoupling of dmPFC-PVI activity preceding an active bout, accompanied by a decrease in the behavioral time spent in active interactions (Fig. 4). Failed activation of adult jSI dmPFC-PVIs prior to initiation of active social bouts may be due to insufficient input drive seen in electrophysiology experiments. These deficits in inputs may in turn disrupt temporal rhythms of dmPFC-PVIs in jSI mice necessary to coordinate downstream excitatory neurons to trigger social approach. To our surprise, patch-clamp recordings did not detect differences between isolated and group housed mice in dmPVI-PFC electrophysiology immediately following isolation (Fig. 5). This suggests that if social signals drive dmPFC-PVI maturation directly, differences at the electrophysiological level are not yet apparent immediately following isolation, and emerge later into adulthood. This phenotype can be described as a developmental "freeze" where dmPFC-PVIs continue to mature between the late juvenile period and adulthood in group housed mice, but this late development is absent in isolated mice. The altered developmental trajectory in dmPFC-PVIs seen in our study may reflect a deficit that emerges during the isolation experience but is initially encoded by molecular changes that may precede electrophysiological changes. Indeed, in the sensory cortex, as PVIs mature, their transcriptional program shifts, leading to expression of channels and other proteins that allow for increased excitability, increased firing rates, and sustained firing, characteristic of the mature fast spiking pattern[41]. Similarly, we hypothesize that lack of social experience during the sensitive window may alter the epigenome, transcriptome or proteome of PV neurons which may prime the cell for later developmental changes.

Impairments in PVI functioning within the brain, particularly in the dmPFC, are a common finding in neurodevelopmental disorders with social deficits. For example, autism patients have fewer PVIs in the dmPFC and schizophrenic patients show a reduction in PV expression, and in levels of the enzyme responsible for synthesizing GABA, glutamic acid decarboxylase (GAD) within PV cells[15–17]. A large body of evidence points toward decreased gamma oscillation power, indicative of decreased PVI activity, in both schizophrenics and autistic patients[18,19]. Our study uncovers a key time window and cell-type for social behavior development, which will inform future research towards behavioral and pharmacological interventions.

## Methods

**Animals.** Male C57Bl/6 wild-type mice, *PV-Cre* mice (stock number 017320, Jackson Laboratory), or PV-GFP mice (*PV-Cre* mice crossed Cre-dependent eGFP-L10a mice (Ribo-GFP #024750, Jackson Laboratory) were used. Animals were group housed in standard laboratory cages in a temperature- and humidity-controlled vivarium with a 12:12 light/dark cycle. Food and water were provided ad libitum throughout the experiment. Viral injections were performed when mice were 9–10 weeks old for adult experiments, and between p42 and p46 for juvenile isolated animals. Behavior experiments took place when mice were 4–10 months old, or 2–4 months old for juvenile isolated mice. For juvenile social isolation (jSI), wild-type C57B6 mice (Charles River Laboratories, shipped at p14, jSI vs GH comparison and Shuffle vs. GH comparison), *PV-Cre* mice (eDREADD experiments, adult isolation) or PV-GFP mice (electrophysiology experiments) were

isolated from weaning (p21) for two weeks. Mice were re-grouped with age-, sex-, and strain-matched jSI males at p35 (3–5 mice per cage), co-housed for 1 month and then behaviorally tested in adulthood, between 2 and 4 months. For the "shuffled" condition, mice were group housed following weaning and were re-grouped with unfamiliar mice (3–5 mice per cage) at p35, co-housed for 1 month and then tested between 2 and 4 months. For the adult isolation condition, mice were isolated for 2 weeks starting between 2 and 4 months, and were then co-housed for 1 month followed by behavior testing. All experiments complied with ethical regulations for animal testing and research. All animal protocols were approved by IACUC at the Icahn School of Medicine at Mount Sinai.

**Viral injection**. For dmPFC injections, mice were anesthetized with isoflurane and head-fixed in a mouse stereotaxic apparatus (Narishige). Bilateral injections were made at three locations per hemisphere: (1) AP + 1.7 mm, ML ± 0.2 mm, and DV—1.0 mm; (2) AP + 1.1 mm, ML ± 0.2 mm, and DV—0.80 mm; (3) AP + 0.4, ML ± 0.2 mm, and DV—0.75 mm. Each injection contained 500 nl of AAV8-hSyn-DIO-hM4Di-mCherry (iDREADD) (University of North Carolina Vector Core) or AAV8-hSyn-DIO-hM3Dq-mCherry (eDREADD, addgene) infused at a rate of 200 nl/min using a microinjector and 2.5-μl Hamilton syringe. For Fiber Photometry Surgeries, a unilateral injection was made at AP 1.1, ML—0.2 mm, and DV—0.8 mm with 500 nl of AAV1-Syn-FLEX-GCaMP6f-WPRE-SV40 (Penn Vector Core, addgene). A 1.3-length mm fiber-optic cannula with a 0.48 numerical aperture and a 400 -μm core diameter (Doric Lenses) was implanted over the injection site using metabond and dental cement. For wireless optogenetics surgeries, a 1-mm diameter hole was drilled bilaterally from 0.7 mm AP to 1.7 mm AP, and injections of 500 nl of AAV1-EF1-DIO-hChR2-mCherry (addgene) were made at: AP + 1.7 mm, ML ± 0.2 mm, and DV—1.0 mm; (2) AP + 1.1 mm, ML ± 0.2 mm, and DV— 0.80 mm; (3) AP + 0.7, ML ± 0.2 mm, and DV—0.75 mm. A 1.1 mm length, 1.2 mm distance center to center, 500-μm diameter LED optic fiber (Amuza) was then implanted at the center (1.1 mm) of the skull hole.

**Behavior testing**. For behavior testing of iDREADD and eDREADD manipulations, Clozapine-N-oxide (CNO; Tocris Bioscience) was fully dissolved in saline and injected intraperitoneally (i.p.) at a 10 mg/kg dose (iDREADD experiments) or 1 mg/kg (eDREADD experiments) 30 min before behavior testing. Mice received CNO and saline (SAL) in a counterbalanced fashion, or in optogenetic experiments, received either light or no light with at least 5 days of time between. Behavior testing order was social behavior (e.g., 3-chamber test and/or reciprocal interactions test) prior to any control behaviors, such as open field or anxiety behavior. eDREADD mice were run in two separate cohorts, one for social behavior, and a second cohort for anxiety profiling behavior. Mice were excluded from behavior tests if they demonstrated significant alterations in motor responses due to i.p. procedures.

Reciprocal interactions: To test a larger and more naturalistic array of social behaviors between two freely interacting mice, we turned to a reciprocal interactions test. We tested mice for 5 min with a 5 min baseline in an open-field apparatus (43 cm × 43 cm × 33 cm) while videotaping, and investigators blinded to experimental conditions scored eight different commonly observed behaviors including anogenital sniffing (ag), nose-to-nose (n2n), anogenital sniffing while following the stimulus mouse (ag/follow), following (follow), orienting to the stimulus mouse (orient), approaching the stimulus mouse (approach), investigation of the stimulus mouse (investigation), and being investigated by the stimulus mouse (investigated by stim), adapted from refs. [42,43]. Rare behaviors, such as being chased or chasing, grooming, or side-by-side huddling were also scored but then were excluded due to very low frequencies. Behaviors were then grouped into "active" "passive", "mutual", or "orient" categories for downstream analyses. Active behavioral bouts were defined as bouts initiated by the focal mouse. These bouts included behavior strings of sub-behaviors within the "active" group, including the focal mouse orienting to the stimulus, followed by approach, etc. Orient bouts that lasted longer than 5 s without being followed by another "active" sub-behavior were considered on their own as part of the "orient" category. Active bouts also included "ag", "n2n", "ag/follow", "follow", and "investigation". These behaviors were considered part of the same bout if they occurred with less than 3 s between the end of one and the start of another. The start of an active bout was defined as the initiation of the first behavior in a string of active behaviors, with <3 s between them. Passive bouts included "investigated by stim". The initiation of the passive bout was defined as when the stimulus mouse began being investigating the focal mouse, but does not include the approach time of the stimulus mouse. Mutual bouts are defined as bouts where the behavioral scorer could not determine which mouse initiated the behavioral encounter. These bouts were scored but then excluded from downstream analysis due to the relative infrequency. In order to examine the distinct behavioral sequences and frequency of transitions between behaviors across groups, the sequence of behaviors was transformed into transition matrices for each group using R scripts and diagrams depicting frequency of transitions between behaviors were visualized using circos[44] (see "Statistics" sections for statistical comparisons).

Three-chamber test: Testing was conducted in the sociability cage (Noldus), a 3-chambered rectangle with clear acrylic walls and a white matte bottom. Mice were habituated to the center chamber for 10 mins, followed by a 10 min habituation to the full chamber one day prior to testing. On the day of testing, mice were placed in the center chamber for 5 min, followed by 5 min exploration of the full chamber with wire corrals. To test sociability, an age-, sex-, and strain-matched mouse was placed under a wire corral in the "social chamber", and a novel object was placed under the corral in the "object chamber". The subject mouse was allowed to freely investigate for a 10-min test phase. Behavior was recorded and scored by Ethovision (Noldus: v.9,14). "Social interaction time" or "object interaction time" was defined as the nose point (detected by Ethovision) in the interaction zone, which is a circular zone surrounding the corral containing the mouse or object.

Open field: To assess anxiety and locomotor behaviors, we used a square acrylic apparatus (43 cm × 43 cm × 33 cm) equipped with a panel of 16 horizontal infrared beams per axis. Data were collected and characterized with Fusion v4 software (Omnitech Electronics). Exploratory behavior and locomotor activity were measured for 30 min. We collected data for time spent in center vs. periphery, distance traveled, rears, and stereotypic behaviors.

Light–dark box: To assess anxiety related to exploring bright-aversive environments, we used the light–dark box[45]. An open-field square acrylic box (43 cm × 43 cm × 33 cm) equipped with a panel of 16 horizontal infrared beams per axis was divided into two equal zones, one covered by a dark acrylic enclosure with holes on all four sides to allow for beam brakes to continuously track the mouse during behavioral testing and with an animal entrance opening (10 cm × 3.2 cm, Omnitech Electronics). The mouse was started in the light portion of the testing arena, and time in the light and dark sides of the box were collected during a 10-min behavioral testing session.

Elevated-plus maze: The elevated-plus maze is a frequently used behavioral assay to investigate anxiety and exploratory behavior by comparing time spent in enclosed arms compared with open arms in a plus-shaped maze with four equal size arms that are elevated above the ground[46,47]. Mice were placed in the center chamber and allowed to freely explore for 8 min. Behavior was recorded and scored by Ethovision (v.14).

**Fiber photometry**. Imaging: Experiments were done at least 3 weeks after viral injection to allow for sufficient viral expression. During recording sessions, two excitation LEDs (Thor Labs) reflected off dichroic mirrors to record GCaMP6f specific signal (465) and nonspecific autofluorescence-related signals as a control (405). A fiber-optic patch cord (Doric Lenses, MFP_400/430/0.48_1.3) was attached to the implanted fiber-optic cannula with cubic zirconia sheaths. The fiber-optic cable was coupled to two LEDs, which pass light through a GFP or violet excitation filter (Thor labs) and dichroic mirrors. Emitted light recorded from the brain through the fiber-optic cable is passed through the dichroic mirror and emission filters, and through a 0.50 N.A. microscope lens (62-561, Edmund Optics) and then focused and projected onto a photodetector (Model 2151 Femtowatt Photoreceiver). A real-time signal processing (RX8, Tucker-Davis Technologies) software designed using OpenEx was used to sinusoidally modulate each LED's output at different frequencies to un-mix signals from each LED. Signals were collected at a sampling frequency of 381 Hz (reciprocal interactions). For 3-chamber photometry studies, a new system was used to acquire the data, including two excitation lasers 465 and 405 (Doric) with an approximate average light of 18 μW. We used an updated digital fiber photometry processor (RZ5P, Tucker-Davis Technologies). Signal was pre-processed using the Synapse Software Suite (Tucker-D Technologies) and collected at a sampling frequency of 1018 Hz.

Reciprocal interactions test behavior: Prior to behavior testing, mice were habituated during three 30 min sessions to the fiber-optic cable. For analysis of activity during social exploration and object exploration, a baseline was recorded for each mouse in a novel open-field arena at the start of testing. Following baseline, we placed a novel object or a novel age-, sex-, and strain-matched mouse in the arena (counterbalanced between object and social) for a 5 min trial-specific behaviors were scored, as described above.

Three-chamber test behavior: 3-chamber behavior testing was conducted as described above, for photometry analysis, 3-chamber entries were only considered if the mouse was in the chamber for >1 s, and chamber entries were annealed together if there was <1 s between them. TTLs were sent from ethovision to align behavioral traces with photometry signal, using the mini-USB input/output box (Noldus).

Analysis: Analysis was performed using custom python scripts. Each channel was first normalized by subtracting the median voltage for the entire recording period for both the signal (465) and the noise (405). The 405 channel was then normalized using an sgolay filter to remove high-frequency changes in the 405 channel, and then subtracted from the 465 channel. This served as a control for movement related signals and fluorescence bleaching. The signal channel was then z-scored using a 30 s baseline directly before adding a mouse or an object for reciprocal interactions photometry experiments, or was z-scored to the entire 10 min behavioral test for 3-chamber. To compare social vs. object dmPFC-PVI activity in the reciprocal interactions test, we compared mean z-score during baseline periods (30 s before introducing a stimulus) compared with the 30 s after introducing either a social stimulus or an object. To compare dmPFC-PVI activity before and after active and passive bouts in the reciprocal interactions test, and to compare social chamber entries in the 3-chamber test, each trace was additionally normalized with a linear detrend. This normalization was only used to compare shorter GCaMP activation, on the order of a few seconds, because this normalization preserved higher frequency changes seen before active bouts, but did

not preserve larger scale changes seen when comparing across pre and post social stimulus.

**Wireless optogenetics**. Behavior: Prior to behavior testing, mice were habituated to either the 3-chamber or the open field for one 20 min habituation phase while wearing a dummy version of the receiver (Amuza, Teleopto wireless optogenetics system). 3-chamber: Mice were habituated to the center chamber for 5 min and the whole chamber for 5 min, then mice were presented with a novel social target and a novel object for a 10 min test phase. During the 10 min test phase, mice were tested in a 2 min "on" 2 min "off" counterbalanced protocol. During the "on" phase, 3 s of 40 Hz light delivered at a 5% duty cycle to minimize heat increases in the brain, potentially leading to nonspecific effects of light activation, seen when higher duty cycles are used (Amuza) was triggered when mice were in the center zone. Chamber entries (social chamber entry, object chamber entry, or remain in center chamber) within 5 s of the end of the pulse (or within 8 s of the start of the pulse) were measured and compared in the light "on" and light "off" conditions. Open Field and reciprocal interactions: Mice were placed in the open field for a 5-min test session, followed by a 5 min reciprocal interaction session with a novel age-, sex-, and strain-matched mouse. Three seconds 40 Hz pulses were randomly generated and spaced 8 to 15 s apart. Velocity (for open field) or social behaviors (see "Behavior" section for details) were analyzed in the 8 s following the pulse initiation.

In vivo electrophysiological validation: Sixteen-channel silicone probes with $177\,\mu m^2$ recording sites (NeuroNexus Technologies) spaced $50\,\mu m$ apart were used to record neuronal activity in the PFC. All in vivo recordings were acquired using the Omniplex system (Plexon). Spike signals were filtered at a bandpass of 300 Hz to 8 kHz. Sorting of single units was carried out using principal component analysis in an offline sorter (Plexon). Three weeks after virus injection, electrophysiological experiments with optogenetic stimulation were performed under 0.8–1% isoflurane anesthesia. Laser (wavelength 473 nm, 1 ms duration, 40 Hz) was delivered using an optic fiber (diameter $105\,\mu m$) coupled to the extracellular recording electrode. The power at the fiber-optic tip was ~10 mW.

**Cortisol**. JSI and GH mice were rapidly decapitated at either p35 or > p60 between 2 and 4 h before the start of the dark cycle. Whole trunk blood was collected, and serum was extracted by allowing samples to clot at room temperature for 30 min and then spinning at $5000 \times g$ to separate serum from blood cells. Serum corticosterone concentrations were obtained by running a corticosterone competitive ELISA assay (Invitrogen) followed by fitting to a standard curve (CurveExpertPro).

**Patch-clamp recording from dmPFC-PVIs**. Animals were decapitated under isoflurane anesthesia. Brains were quickly removed and transferred into ice-cold (0–4 °C) artificial cerebrospinal fluid (ACSF) of the following composition (in mM): 210.3 sucrose, 11 glucose, 2.5 KCl, 1 $NaH_2PO_4$, 26.2 $NaHCO_3$, 0.5 $CaCl_2$, and 4 $MgCl_2$. Acute coronal slices of ACA (350 μm) contained both hemispheres. Slices were allowed to recover for 40 min at room temperature in the same solution, but with reduced sucrose (105.2 mM) and addition of NaCl (109.5 mM). Following recovery, slices were maintained at room temperature in standard ACSF composed of the following (in mM): 119 NaCl, 2.5 KCl, 1 $NaH_2PO_4$, 26.2 $NaHCO_3$, 11 glucose, 2 $CaCl_2$, and 2 $MgCl_2$. Patch-clamp recordings were performed from fluorescently labeled PVI neurons in deep layer dmPFC (including anterior cingulated cortex and prelimbic cortex between AP 2.3 mm to 1.5 mm) using borosilicate glass electrodes (3–5 MΩ). Whole-cell voltage-clamp recordings were obtained with the internal solution containing (in mM): 120 Cs-methanesulfonate, 10 HEPES, 0.5 EGTA, 8 NaCl, 4 Mg-ATP, 1 QX-314, 10 Na-phosphocreatine, and 0.4 Na-GTP. Current-clamp recordings were obtained with the internal solution containing (in mM): 127.5 K-methanesulfonate, 10 HEPES, 5 KCl, 5 Na-phosphocreatine, 2 $MgCl_2$, 2 Mg-ATP, 0.6 EGTA, and 0.3 Na-GTP. Spontaneous excitatory and inhibitory postsynaptic currents were conducted in standard ACSF (as above). Miniature excitatory and inhibitory postsynaptic currents (mEPSC and mIPSC) were recorded in the presence of TTX (1 μM; Abcam) in the bath solution. Spontaneous and Miniature EPSCs and IPSCs were separated by holding the neuron at the reversal potential for excitatory or inhibitory postsynaptic currents, allowing for isolation of EPSCs and IPSCs at −60 mV and 0 mV, respectively. Data were low-pass filtered at 3 kHz, and acquired at 10 kHz using Multiclamp 700B (Axon Instruments) and pClamp 10 (Molecular Devices). Series and membrane resistance was continuously monitored, and recordings were discarded when these measurements changed by > 20%. Recordings in which series resistance exceeded 25 MΩ were rejected. Detection and analysis of EPSCs and IPSCs were performed using MiniAnalysis (Synaptosoft). For current-clamp recordings, we applied DNQX (20 μM; TOCRIS), D-AP5 (50 μM; TOCRIS), and picrotoxin (30 μM; TOCRIS). Series resistance was monitored and canceled using a bridge circuit, and pipette capacitance was compensated. Voltage signals were low-pass filtered at 10 kHz. The baseline membrane potential was maintained near −70 mV with a current injection. We recorded membrane potential responses to hyperpolarizing and depolarizing current pulses (500 ms in duration) and then, we examined action potential by using Signal 4 (Cambridge Electronic Design).

**Immunohistochemistry**. Mice were anesthetized with isoflurane and transcardially perfused with cold 0.1 M phosphate buffer, followed by 4% paraformaldehyde (PFA) dissolved in 0.1 M PB. For eDREADD validation by egr-1 immunostaining, mice were perfused 120 min after the injections of either Clozapine-N-oxide or saline. Brains were post-fixed between 6 and 12 h in 4% PFA at 4 °C, and cryoprotected in 30% sucrose. Frozen brains were sectioned into 35 μM sections using a cryostat (CM3050, Leica). Free floating sections from whole brain were washed 3× in tris-buffered saline (TBS) and then blocked in 1% bovine serum albumin (BSA) in TBST (0.25% Triton X-100 in TBS) for 1 h. Sections were incubated overnight at room temperature in primary antibody (rabbit anti-egr-1 (Santa Cruz, cat#: sc–189, 1:10,000) mouse anti-parvalbumin (1:500, cat#: 235, Swant, Switzerland), or rabbit anti-parvalbumin (1:1000, cat#: PV27, Swant). Slices were then washed in 1% BSA in TBST, followed by incubation with appropriate secondary antibodies (Alexa 647-conjugated donkey anti-mouse cat# A-31571, Alex 568 goat anti-rabbit (cat# A-11036), or Alexa 647 goat anti-rabbit (cat# A-21245) (All 1:200, Life Technologies)). Sections were then mounted with DAPI Fluoromount-G (Southern Biotech).

**Imaging and quantification**. Imaging was performed using LSM780 confocal microscopes (Zeiss) and processed and analyzed using ImageJ32 software (NIH). For viral expression analysis and egr-1 quantifications, images were background subtracted and then thresholded using the Max Entropy function. Averages were collected from three to five brain slices per mouse for iDREADD and eDREADD validation. For wireless optogenetics validation, LED placement was confirmed in all mice. For photometry validation, we confirmed signal levels in all animals prior to behavior testing (a change in signal all least tenfold higher than the signal change in the autofluorescence channel). After behavior testing, we confirmed ferrule placement for a subset of mice (all mice shown in Figs. 1 and 4). Co-localization analysis to determine the number of PV cells co-expressing egr-1 or the number of mCherry-positive cells co-expressing PV stain was performed using the analyze particles function in each channel, followed by counting the intersection of the particles for each region of interest.

**Electrophysiological validation of iDREADD**. To validate suppression of dmPFC-PVIs with iDREADD, male C57Bl/6 *PV-Cre* mice were injected with AAV8-DIO-hM4Di-mCherry and recorded 3 weeks after injections. Mice were anesthetized with isoflurane, decapitated, and their brains were quickly removed. Coronal slices (350 μm) containing the dmPFC were sectioned using a VT1200S vibratome (Leica Microsystems, Buffalo Grove, IL, USA) in sucrose dissection solution (in mM: 210.3 sucrose, 26.2 $NaHCO_3$, 11 glucose, 4 $MgCl_2$, 2.5 KCl, 1 $NaH_2PO_4$, 0.5 ascorbate, and 0.5 $CaCl_2$) chilled to −4 °C. Slices were recovered in standard ACSF (in mM: 119 NaCl, 26.2 $NaHCO_3$, 11 glucose, 2.5 KCl, 2 $CaCl_2$, 2 $MgCl_2$, and 1 $NaH_2PO_4$) for 40 min at 34 °C, and then were maintained at room temperature for the remainder of the recording. All solutions were continuously bubbled with 95% $O_2$:5% $CO_2$. Whole-cell recordings in gap-free mode were obtained with borosilicate glass electrodes (5–8 mΩ resistance) filled with a current-clamp internal solution (in mM: 127.5 K-methanesulfonate, 10 HEPES, 5 KCl, 5 Na-phosphocreatine, 2 $MgCl_2$, 2 Mg-ATP, 0.6 EGTA, and 0.3 Na-GTP). PVIs expressing mCherry in the dmPFC region were visualized on an upright microscope equipped for both DIC and fluorescence visualization. Cells were held at −55 mV. Clozapine-N-oxide (CNO; Tocris Bioscience 10 μM) was bath applied to cells that exhibited at least 5 min of stable baseline recording. Resting membrane potential was compared between baseline and 5 min after CNO application.

**Statistics**. Statistical analyses were performed using Prism (Graphpad), SPSS, and R. Statistical analyses for fiber photometry were conducted using parametric tests on z-scored data following tests for normality. Data that did not meet normality criteria were assessed using non-parametric tests. Analyses comparing multiple time points were conducted with one-way repeated measures ANOVAs or paired t tests as indicated. For behavioral sequence analysis, transition matrices for frequency of behavioral sequence transitions were created and transformed to find joint probability matrices. These matrices were then statistically compared by running 10,000 permutations on shuffled data for both treatments to create a null distribution. We also ran left and right permutation tests to assess transitions from and to specific behaviors. Wireless optogenetic experiments were analyzed using linear mixed models in R (lsmeans package), pulse number and animal number as random factors, and treatment (on vs. off) and behavior category (active, passive, orient) as fixed factors. Post hoc tests were conducted with difflsmeans. Behaviors presented at the animal level were analyzed with paired t tests. Behavioral tests in DREADD experiments were analyzed using repeated measures ANOVAs followed by post hoc tests using Bonferroni's multiple comparison corrections, or Student's t tests as indicated. For PVI patch-clamp physiology, statistical analyses were conducted using two-way ANOVAs and planned post hoc tests. All statistical tests were two-tailed. Bar graphs and photometry averaged traces are presented as the mean and error bars represent the standard error of the mean (SEM).

**Reporting summary**. Further information on research design is available in the Nature Research Reporting Summary linked to this article.

## Data availability

The source data underlying Figs. 1–6 and Supplementary Figs. S1–S17 are provided as a source data file. The rest of relevant data not included in the source data file are available from the authors upon request.

## Code availability

All scripts used to analyze or display the data are available upon request.

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

## Acknowledgements

This work was supported by National Institute of Health T32MH966785 to L.B., Naito Foundation, Uehara Memorial Foundation, Mochida Memorial Foundation, JSPS to K.Y., R01MH118297, R01MH119523, and the Simons Foundation Autism Research Initiative  to H.M and R01MH104341, R21NS097664, P50MH096890 subproject ID: 8478 to S.A., and the Brain Research Foundation to H.M. and S.A.

## Author contributions

L.B., H.M., and S.A. designed and analyzed all experiments and wrote the paper. L.B. performed surgeries and behavior for all experiments in part assisted by H.K. D.K. performed in vivo electrophysiology experiment. K.Y. performed patch physiology experiments. E.K.L. and R.C. performed patch physiology experiments for iDREADD validation. L.B., M.F., D.B., and S.R. performed fiber photometry experiments and analysis. M.P. and J.K. assisted with viral quantification and behavior scoring. S.C. and D.B. provided scripts for behavioral transition data, and M.S. provided scripts for linear mixed modeling analysis. K.N. provided instruction and support for wireless optogenetics surgery and implementation.

## Competing interests

The authors declare no competing interests.

**Additional information**

