## [Peer Review File · Nature Communications]

Reviewers' comments:

Reviewer #1 (Remarks to the Author):

In this interesting and timely report, Bicks et al. use fiber photometry, optogenetics, chemogenetics, ephys, and a variety of clever approaches to analyzing social behavior to show that PV interneurons in the medial prefrontal cortex play a critical role in initiating active social approach behaviors. They also show that juvenile social isolation disrupts the functional development of prefrontal PV interneurons and that chemogenetically increasing their excitability in adulthood is sufficient to rescue social behavior deficits in adulthood caused by social isolation early in life.

Overall, this is a very interesting paper on a topic of special interest to many investigators in the field. The experiments are well executed and thoughtfully interpreted, and I especially appreciated the various controls incorporated into many of the key conclusions.

I have a few suggestions that could strengthen this work prior to publication:

1. Arguably the most interesting and novel results in this paper come from the juvenile social isolation experiments. The chemogenetic activation experiments in Fig. 6 show convincingly that activating PV cells is sufficient to rescue the effects of jSI on social interaction behavior in adulthood. However, the fiber photometry data in Fig. 4 are critical for showing that PV cell activity is altered by jSI, and they are harder to interpret for several reasons. First, the data in Fig. 4d are interpreted as indicating that PV cell activity is not modulated by active social interactions, but it's hard to be confident that they aren't responding at all because we're only examining a single (initial) behavioral event from each mouse (if I understand correctly -- this part was a little unclear). Consequently, everything hinges on the assumption that the timing of the initiation of the active social behavior can be precisely quantified. Any errors in estimating the onset of these events could lead to offsets in the relative timing of the PV cell response across mice. Some traces in Figures 4 and 1 seem to suggest there are multiple small peaks occurring at different times with respect to time zero, and errors in quantifying the timing could account for this. Apologies if I missed it, but can the authors provide a little more detail on how these events were scored and how confident they are in their timing? Since six different types of behavior are scored as "active", it's also unclear whether the averages represent a single behavior from each mouse or multiple active behaviors (e.g. for a given mouse, the average of the first approach, the first n2n, and the first follow). Please specify what is being averaged in the legend? And a minor point: the denominator degrees of freedom are missing from the ANOVAs in the legend.

2. Similarly, the data in Fig. 4d, right panel are interpreted as indicating that passive interaction behaviors are associated with a decrease in PV activity in jSI mice but not in GH mice. The same question about timing pertains here. However, a potentially bigger problem is that this conclusion assumes that what's happening behaviorally in jSI mice prior to passive social interactions is on average the same as what's happening in GH mice — an apples and apples comparison. But the differing transition states present in fig. 4b suggest the opposite — that it's possible that on average, jSI mice tend to be engaged in a different kind of behavior prior to passive interactions, in which case this could explain the decrease seen in jSI mice prior to passive interaction onset. Can the authors control for this statistically or experimentally?

3. Also curious: was the finding from figure 1 regarding a PV activity effect specific to the first active or passive social interaction replicated in figure 4? That is, Fig. 4 replicates the effect involving initial interactions but does not show whether there are no effects during subsequent interactions, as in Fig. 1. In general, it would be useful for readers to plot responses to each social interaction and the average for all in each category, just for reference in supplementary material.

4. Overall the paper was very well written but I felt that the discussion could be shortened. At the same time, this section could benefit from an expanded discussion of closely related studies that are either not cited or only briefly mentioned. Perhaps most importantly, Reference 21 showed fairly convincingly that PV cell activity is required for normal social interaction behavior, and it might be helpful to revise the introduction and discussion to clarify what was already known and what's new here. It might also be worth discussing these results in the context of work from Hailan Hu's lab on the role of mPFC in social interaction behavior. Also, Kingsbury et al., Cell, 2019.

5. The medial prefrontal cortex is a large and functionally heterogeneous brain region. There are many studies suggesting that dorsal and ventral areas of the mPFC may mediate distinct functions. While I appreciate that the histology might not support claims about a specific mPFC subregion, it might be helpful to differentiate between dorsal and ventral — it looks like all of their manipulations and recordings involve dorsal mPFC?

Minor:

- The results in Figs. 3 and 6, in which PV cells are bi-directionally manipulated, and contrasting behavioral effects are presented are critical and fairly persuasive. However they seem to be analyzed differently and are definitely presented differently. Why not analyze and present them in the same way to make it easier to compare?
- In figure panels presenting overlapping fiber photometry traces (e.g. Fig. 4d), consider rendering the SEM bands semi-transparent so that both traces could be seen more clearly. As is, the underlying trace is hard to see compared to the top trace.
- Consider including some rationale for the duty cycle selected for the optogenetic experiments.
- Consider avoiding the anthropomorphic term "loneliness" when referring to mouse behavior.

Reviewer #2 (Remarks to the Author):

Scientific quality: The study is designed and carried out competently. The methods were employed and described adequately. The introduction and discussion parts were informative and written coherent. The figures reflect the results of the study. The literature data are appropriately presented and correctly interpreted. It was a pleasure to read this manuscript.

Novelty: The results present new information about specific activation pattern of parvalbumin-positive interneurons in the medial PFC only prior to an active bout, or a bout initiated by the focal mouse, but not during a passive bout. The present results indicated the role of social experience-dependent maturation of mPFC-PVI with long-term impacts on social behavior.

Specific comments in relation to this manuscript include:

Under the Methods section, the authors explained that the animals were anesthetized with

isoflurane. Does this have any effect on measured parameters?

Did the authors do the 6-8 independent experiments for ELISA assay for the quantification of corticosterone in serum or it was pulled?

Regarding corticosterone assay, did all samples measure in duplicate or triplicate in one assay? What was the variation between samples in corticosterone assay?

Were all the behavioral tests done on the same day? How did the authors make sure, that earlier tests (in which they found differences) did not confound later ones? Most importantly, all behavioral experiments seem to be done in a status of 24 h water and food deprivation. How can the authors exclude that this may have influenced the behavioral read-outs?

Authors should also describe how much time elapsed between the different behavioral tests.

How much slices from each mice brain in the group authors used for immunofluorescence staining?

Specify if an assessment of the normality of data was carried out.

Reviewer #3 (Remarks to the Author):

By using comprehensive and elegant combinations of in vivo fiber photometry imaging, patch-clamp recordings, optogenetics, chemogenetics and behavior approaches, the authors have clearly demonstrated the essential role of prefrontal (mPFC) parvalbumin interneurons (PVIs) in active social approach behavior of mice. They show that 1) mPFC-PVIs are activated prior to an active social approach towards another mouse, 2) brief optogenetic activation of mPFC-PVIs promotes social approach, and 3) suppressing mPFC-PVIs decreases social interaction but have no impact on exploration, anxiety.

Although the link between mPFC-PVIs and social behavior have been recently reported, the present study provides an important novel neurodevelopmental insight. They show that social isolation during a two-week window at juvenile age (P21-35) (but not in adulthood) leads to social deficits in adult mice. Juvenile social isolation also altered mPFC-PVIs activity during social exploration. This social isolation induced altered activity of mPFC-PVIs comes with a decreased excitability of these interneurons and reduced synaptic input drive. These physiological anomalies of mPFC-PVIs in adulthood were however not observed in young mice just following the 2-week isolation period. This indicated that social isolation in juvenile leads to a disruption of normal late maturation of mPFC-PVIs and some neuronal networks which feed into these mPFC-PVIs. This also suggests that juvenile age may constitute a critical period of plasticity in prefrontal cortex in relation to social behavior. Moreover, the authors demonstrated that enhancing the activity of mPFC-PVIs (using excitatory DREADDS expressed in mPFC-PVIs) in adult mice who have been isolated during juvenile age can rescue the deficit. This elegantly demonstrates again the importance of these interneurons and provides an interesting framework to develop novel strategy to ameliorate social behaviors in schizophrenia and autistic patients.

This is an innovative and exciting paper which reveals important insights linking critical windows for social experience with mPFC local circuit maturation in control of social behaviour, a transdiagnostic dimension which are impaired in various neurodevelopmental and psychiatric disorders. This knowledge thus has a high translational value.

The experimental approach is solid, and the presented data are novel and convincing. The manuscript does not address experimentally which network(s) of neurons connected to mPFC-PVIs are involved in altering social behavior following an early-life social isolation. However, the authors provide a quite complete discussion regarding this aspect, which should be further investigated in the future.

Following are some specific minor comments:

- Lines 403-405: "This altered synaptic drive could be due to increased activity of somatostatin interneurons, leading to inhibition of PVIs, or may be related to reduced direct activating synaptic inputs, such as those arising from the medial dorsal thalamus, which have been shown to regulate social behavior": If there was an increased activity of somatostatin interneurons, should we observe an increase in IPSPs in isolated mice ? (which is not the case!).
- Lines 456-459: what could be the mechanism underlying the absence of increased excitability following juvenile isolation? Epigenetic is likely a player, what candidate channel(s) could be the target of epigenetic mechanisms? Please discuss
- Figure 6e: It looks like there might be an interaction effect between housing and drug in the light/dark test. There is no mention about it. Was the interaction tested?
- Legend figure7: the legend is not very clear. For instance:
 - "short PVI activation leads to social approach": it is not clear whether in this particular case the authors describe the neuronal activity observed during the behavior or the activity induced optogenetically or chemogenetically.
 - "leads to social approach, but not in juvenile social isolated mice": Is it adult mice that have been isolated during juvenile age ?
 - "social interaction of isolated mice can be mitigated by chemogenetically increasing PFC-PVI activity"

I would also strongly advice a careful editing and correction of many typing errors.

- Line135: "post (0 second to -1 second)": is it not "0 to 1 second" ?
- Line 172: "mice made significantly more entries into the social chamber, during the 'on' phase.": the statistic seems to be missing
- Line 268: "due to lack of social-dependent activation of mPFC-PVIs"
- Lines 286-288: "sEPSCs and sIPSCs showed a developmental decrease in frequency as well as a significant main effect of age housing, showing a decrease in sEPSC frequency between jSI and GH (Fig. 5c, 288 left: 2-way ANOVA, age factor: $p < 0.001$, housing factor: $p = 0.01$)."
- Line 350: "shared across of a range of disorders"
- Line 421: "social isolation for two weeks does not impact does not impact adult social behavior"
- Line 743: "reveals significant differences between the joint distribution matrix between of jSI and GH mice"
- Line 752: "Change in z-score from baseline in the first active 753 encounter shows a significant difference"
- Line 895...: where is the mouse placed at the beginning of the Light/dark test ?
- Line 929-931: "To compare social vs. object mPFC-PVI activity, we compared mean z-score during baseline periods (30 sec before introducing a stimulus) compared with the 30 sec after introducing either a 931 social stimulus or an object."
- Line 943-44: "If mice stayed in the center, Chamber entries (social chamber entry, object chamber entry, or remain in center chamber) within 5 seconds of the end of the pulse...": Sentence not very understandable sentence. If the mouse stays in the center, we cannot measure entries...!
- Line 975: anterior cingulate cortex

Reviewer #1:

In this interesting and timely report, Bicks et al. use fiber photometry, optogenetics, chemogenetics, ephys, and a variety of clever approaches to analyzing social behavior to show that PV interneurons in the medial prefrontal cortex play a critical role in initiating active social approach behaviors. They also show that juvenile social isolation disrupts the functional development of prefrontal PV interneurons and that chemogenetically increasing their excitability in adulthood is sufficient to rescue social behavior deficits in adulthood caused by social isolation early in life.

Overall, this is a very interesting paper on a topic of special interest to many investigators in the field. The experiments are well executed and thoughtfully interpreted, and I especially appreciated the various controls incorporated into many of the key conclusions.

I have a few suggestions that could strengthen this work prior to publication:

1. Arguably the most interesting and novel results in this paper come from the juvenile social isolation experiments. The chemogenetic activation experiments in Fig. 6 show convincingly that activating PV cells is sufficient to rescue the effects of jSI on social interaction behavior in adulthood. However, the fiber photometry data in Fig. 4 are critical for showing that PV cell activity is altered by jSI, and they are harder to interpret for several reasons. First, the data in Fig. 4d are interpreted as indicating that PV cell activity is not modulated by active social interactions, but it's hard to be confident that they aren't responding at all because we're only examining a single (initial) behavioral event from each mouse (if I understand correctly -- this part was a little unclear). Consequently, everything hinges on the assumption that the timing of the initiation of the active social behavior can be precisely quantified. Any errors in estimating the onset of these events could lead to offsets in the relative timing of the PV cell response across mice. Some traces in Figures 4 and 1 seem to suggest there are multiple small peaks occurring at different times with respect to time zero, and errors in quantifying the timing could account for this. Apologies if I missed it, but can the authors provide a little more detail on how these events were scored and how confident they are in their timing? Since six different types of behavior are scored as "active", it's also unclear whether the averages represent a single behavior from each mouse or multiple active behaviors (e.g. for a given mouse, the average of the first approach, the first n2n, and the first follow). Please specify what is being averaged in the legend? And a minor point: the denominator degrees of freedom are missing from the ANOVAs in the legend.

Response: We thank the reviewer for this important feedback. In our revised manuscript, we have added a new photometry cohort using a separate behavioral paradigm, the 3-chamber test, to add more objective and precise time measurements and to confirm our findings in a convergent test (Fig. S14). Based on our results showing a pre-Active behavior increase in PFC-PVI activity, we hypothesized that in the 3-chamber test, we would see increased PFC-PVI activity prior to entering the social chamber in GH, but not in jSI mice. Indeed, in our revised manuscript we now show that across all social chamber entries, there is significantly higher activity in the 1 second prior to the social chamber entry in GH compared with jSI mice but no difference in activity following social chamber entry (Fig. S14). This measure is precise in that we aligned the ethovision signal to the Photometry signal using TTLs, and used data from ethovision to get a precise, sub-second timestamp for when the nose-point of the mouse crosses into the social chamber. This new analysis, in a separate cohort, confirms pre-active social behavior PFC-PVI activity in GH, but not jSI mice. We have also added more detail in the methods to address the reviewers point that the scoring of the events and the specific behaviors that were averaged was not clear. To clarify here, the averages represent a single active bout, which is composed of string of behaviors that fall within the active category that are closely linked together in sequence (<3 seconds apart). We plotted starting with the initiation of whichever first behavior which belongs to "active" category, for example, orienting to the stimulus, if followed by approach, etc. Passive bouts start when the stimulus mouse begins investigating the focal mouse, if the investigation is passive (i.e. the focal mouse does not actively engage in any investigation behaviors). Behaviors that could not be attributed as active or passive (i.e. mutual), were scored but dropped from this analysis. Regarding the statistics, in our revised manuscript, we now added the denominator degrees of freedom are missing from the ANOVAs in the legend.

2. Similarly, the data in Fig. 4d, right panel are interpreted as indicating that passive interaction behaviors are associated with a decrease in PV activity in jSI mice but not in GH mice. The same question about timing pertains here. However, a potentially bigger problem is that this conclusion assumes that what's happening behaviorally in jSI mice prior to passive social interactions is on average the same as what's happening in GH mice — an apples and apples comparison. But the differing transition states present in fig. 4b suggest the opposite — that it's possible that on average, jSI mice tend to be engaged in a different kind of behavior prior

to passive interactions, in which case this could explain the decrease seen in jSI mice prior to passive interaction onset. Can the authors control for this statistically or experimentally?

Response: *We appreciate reviewer's helpful comments. To address this question, in our initial submission, we used joint probability distributions of transitions between all behaviors, irrespective of the time between the end of behavior 1 and the start of behavior 2. While we found significant differences between the joint probability distributions, indicating an overall difference in the sequence transitions between behaviors, we did not find any specific transition from one behavior to the next that was statistically significant between behaviors, after adjusting for multiple comparisons. We elaborated this point in our revised manuscript.*

To examine the data more in detail, we additionally examined transition matrices to and from each umbrella behavioral state, including just active, passive, and orient states which were grouped for photometry analysis. Specifically, we examine only our overarching behavioral domains, active, passive, and orienting behavior occurring within 3 seconds of each other in order to match the window that was analyzed for photometry signal. This new analysis compares transitions between groups that started at certain behaviors, and transitions between groups that ended at certain behaviors. This would better allow us to determine whether jSI mice might be engaged in a different behavior specifically prior to passive interactions. We found no significant differences between groups in the type of behavior occurring prior to passive behaviors, or any of the behavior categories (active, passive, orient). We also found no differences between groups when examining which behaviors occurred following any behavior category. Therefore, while the joint probability distributions of all transitions are different between groups, when behaviors are grouped into their umbrella categories and only transitions that occur within 3 seconds are examined, we found no differences between groups in transition frequencies starting from or ending with any of our behavior categories. The results of the second analysis are not shown in the text, but are consistent with our initial analysis, making us confident our photometry signal differences are not due to differences in behaviors occurring either prior to or following the active and passive behaviors.

3. Also curious: was the finding from figure 1 regarding a PV activity effect specific to the first active or passive social interaction replicated in figure 4? That is, Fig. 4 replicates the effect involving initial interactions but does not show whether there are no effects during subsequent interactions, as in Fig. 1. In general, it would be useful for readers to plot responses to each social interaction and the average for all in each category, just for reference in supplementary material.

Response: *To address this comment we have added a supplementary figure (Fig.S12) adding the response to each sequential active bout (1st, 2nd, etc). As you can see the first bout is the only bout that shows a significant change from baseline. We also clarify that in Fig. 4 we included GH data already presented in Fig.1 as references so that readers can better compare the data from GH and jSI animals. We made this point clearer in the legend of Fig.4.*

4. Overall the paper was very well written but I felt that the discussion could be shortened. At the same time, this section could benefit from an expanded discussion of closely related studies that are either not cited or only briefly mentioned. Perhaps most importantly, Reference 21 showed fairly convincingly that PV cell activity is required for normal social interaction behavior, and it might be helpful to revise the introduction and discussion to clarify what was already known and what's new here. It might also be worth discussing these results in the context of work from Hailan Hu's lab on the role of mPFC in social interaction behavior. Also, Kingsbury et al., Cell, 2019.

Response: *The reference 21 noted here shows that in the presence of optogenetic stimulation of excitatory neurons in PFC, simultaneous activation of PV neurons is sufficient to ameliorate this deficit. While this convincingly shows that PV cell activity can correct excitatory imbalances to restore social functioning, we feel on its' own this reference does not show that PV cell activity is required for normal social behavior. However, we have revised the introduction to include work from Hailan Hu's lab, which particularly implicates the dmPFC in social behaviors. We also made efforts to shorten the discussion.*

5. The medial prefrontal cortex is a large and functionally heterogeneous brain region. There are many studies suggesting that dorsal and ventral areas of the mPFC may mediate distinct functions. While I appreciate that the histology might not support claims about a specific mPFC subregion, it might be helpful to differentiate between dorsal and ventral — it looks like all of their manipulations and recordings involve dorsal mPFC?

Response: *This is a very helpful point. Indeed, we targeted our injections to include the ACC and the PL, but not the IL which corresponds to dorsal mPFC. In the revised manuscript we have changed the description to reflect that, we now use dmPFC.*

Minor:

- The results in Figs. 3 and 6, in which PV cells are bi-directionally manipulated, and contrasting behavioral effects are presented are critical and fairly persuasive. However they seem to be analyzed differently and are definitely presented differently. Why not analyze and present them in the same way to make it easier to compare?

Response: *While iDREADD did affect social interaction per social chamber entry, it did not abolish social preference, which is now explicitly stated in the text. We now show the social investigation per chamber entry in our iDREADD treated experiments (Fig. 3d) and also show the same analysis for the eDREADD rescue experiments, shown in Fig.S17d for eDREADD rescue.*

- In figure panels presenting overlapping fiber photometry traces (e.g. Fig. 4d), consider rendering the SEM bands semi-transparent so that both traces could be seen more clearly. As is, the underlying trace is hard to see compared to the top trace.

Response: *Thank you for this suggestion, we have revised as you suggested.*

- Consider including some rationale for the duty cycle selected for the optogenetic experiments.

Response: *Thank you for this suggestion. We chosed a lower duty cycle (5%) to minimize the heat increases caused by high power optic stimulation (Amuza manuals). Methods section is revised accordingly.*

- Consider avoiding the anthropomorphic term “loneliness” when referring to mouse behavior.

Response: *We have removed this term.*

Reviewer #2:

Scientific quality: The study is designed and carried out competently. The methods were employed and described adequately. The introduction and discussion parts were informative and written coherent. The figures reflect the results of the study. The literature data are appropriately presented and correctly interpreted. It was a pleasure to read this manuscript.

Novelty: The results present new information about specific activation pattern of parvalbumin-positive interneurons in the medial PFC only prior to an active bout, or a bout initiated by the focal mouse, but not during a passive bout. The present results indicated the role of social experience-dependent maturation of mPFC-PVI with long-term impacts on social behavior.

Specific comments in relation to this manuscript include:

Under the Methods section, the authors explained that the animals were anesthetized with isoflurane. Does this have any effect on measured parameters?

Response: The animals were anesthetized with isoflurane during surgical procedures. This is a common way of conducting animal surgeries for experimental and veterinary purposes. While isoflurane could have some effects (as any anesthetic agent could) our controls and treated animals underwent the same surgery procedure. Therefore we are confident that isoflurane was driving our specific effects.

Did the authors do the 6-8 independent experiments for ELISA assay for the quantification of corticosterone in serum or it was pulled? Regarding corticosterone assay, did all samples measure in duplicate or triplicate in one assay? What *animal variance*.

Response: We have also re-done our analyses to include all three values from each animal in a linear mixed model, using 'sample' as a random factor and treatment as a fixed effect. Results are reported in the figure legend with updated analyses.

Were all the behavioral tests done on the same day? How did the authors make sure, that earlier tests (in which they found differences) did not confound later ones? Most importantly, all behavioral experiments seem to be done in a status of 24 h water and food deprivation. How can the authors exclude that this may have influenced the behavioral read-outs? Authors should also describe how much time elapsed between the different behavioral tests.

Response: Distinct behavior tests were run at least 5 days apart to minimize any confounding effects of one behavior test on the other. However, where possible we did not run the same animal through more than four different behavior paradigms. This is the reason we used a separate cohort to test whether eDREADD manipulations had any effects on anxiety behavior. In this case, the anxiety behaviors were the only behaviors run and we still did not detect an effect. These precautions taken together increase our confidence that there is not a significant anxiety effect of PV manipulations within our paradigm. Animals were not food or water deprived at all in these experiments – we thoroughly checked our methods and made sure to include that animals were given ad lib access to food and water throughout the duration of the study. We now included those details in our method section.

How much slices from each mice brain in the group authors used for immunofluorescence staining?

Response: We used at least three slices per animal for IHC validation of virus. This is included in the methods under the 'Imaging and Quantification' section. We have not included more detail in the section.

Specify if an assessment of the normality of data was carried out.

Response: We did use normality assessments of data and used non parametric tests where necessary. We have now clearly stated this in the methods.

Reviewer #3:

By using comprehensive and elegant combinations of in vivo fiber photometry imaging, patch-clamp recordings, optogenetics, chemogenetics and behavior approaches, the authors have clearly demonstrated the essential role of prefrontal (mPFC) parvalbumin interneurons (PVIs) in active social approach behavior of mice. They show that 1) mPFC-PVIs are activated prior to an active social approach towards another mouse, 2) brief optogenetic activation of mPFC-PVIs promotes social approach, and 3) suppressing mPFC-PVIs decreases social interaction but have no impact on exploration, anxiety.

Although the link between mPFC-PVIs and social behavior have been recently reported, the present study provides an important novel neurodevelopmental insight. They show that social isolation during a two-week window at juvenile age (P21-35) (but not in adulthood) leads to social deficits in adult mice. Juvenile social isolation also altered mPFC-PVIs activity during social exploration. This social isolation induced altered activity of mPFC-PVIs comes with a decreased excitability of these interneurons and reduced synaptic input drive. These physiological anomalies of mPFC-PVIs in adulthood were however not observed in young mice just following the 2-week isolation period. This indicated that social isolation in juvenile leads to a disruption of normal late maturation of mPFC-PVIs and some neuronal networks which feed into these mPFC-PVIs. This also suggests that juvenile age may constitute a critical period of plasticity in prefrontal cortex in relation to social behavior.

Moreover, the authors demonstrated that enhancing the activity of mPFC-PVIs (using excitatory DREADDS expressed in mPFC-PVIs) in adult mice who have been isolated during juvenile age can rescue the deficit. This elegantly demonstrates again the importance of these interneurons and provides an interesting framework to develop novel strategy to ameliorate social behaviors in schizophrenia and autistic patients.

This is an innovative and exciting paper which reveals important insights linking critical windows for social experience with mPFC local circuit maturation in control of social behaviour, a transdiagnostic dimension which are impaired in various neurodevelopmental and psychiatric disorders. This knowledge thus has a high translational value.

The experimental approach is solid, and the presented data are novel and convincing. The manuscript does not address experimentally which network(s) of neurons connected to mPFC-PVIs are involved in altering social behavior following an early-life social isolation. However, the authors provide a quite complete discussion regarding this aspect, which should be further investigated in the future.

Following are some specific minor comments:

- Lines 403-405: "This altered synaptic drive could be due to increased activity of somatostatin interneurons, leading to inhibition of PVIs, or may be related to reduced direct activating synaptic inputs, such as those arising from the medial dorsal thalamus, which have been shown to regulate social behavior": If there was an increased activity of somatostatin interneurons, should we observe an increase in IPSPs in isolated mice ? (which is not the case!).

Response: Thank you for this consideration. We do see a trending interaction between housing and age in the sIPSC frequencies, indicated altered developmental changes and moderately increased sIPSC frequency by adulthood. However, as this increase is subtle and does not meet significance in post hoc tests, we have removed this statement from the discussion

- Lines 456-459: what could be the mechanism underlying the absence of increased excitability following juvenile isolation? Epigenetic is likely a player, what candidate channel(s) could be the target of epigenetic mechanisms? Please discuss

Response: We revised our discussion about epigenetic mechanisms they may alter channel programs in the discussion section.

- Figure 6e: It looks like there might be an interaction effect between housing and drug in the light/dark test. There is no mention about it. Was the interaction tested?

Response: There were no significant interactions detected for any of the anxiety behavior tests, including Light Dark. We now report all interaction F and p values in the main figure legend. Thank you.

- Legend figure7: the legend is not very clear. For instance:

- "short PVI activation leads to social approach": it is not clear whether in this particular case the authors describe the neuronal activity observed during the behavior or the activity induced optogenetically or chemogenetically.
- "leads to social approach, but not in juvenile social isolated mice": Is it adult mice that have been isolated during juvenile age ?
- "social interaction of isolated mice can be mitigated by chemogenetically increasing PFC-PVI activity"

Response: *Thank you for helpful feedback. We have changed this language accordingly.*

I would also strongly advise a careful editing and correction of many typing errors.

- Line135: "post (0 second to -1 second)": is it not "0 to 1 second" ?

- Line 172: "mice made significantly more entries into the social chamber, during the 'on' phase.": the statistic seems to be missing

- Line 268: "due to lack of social-dependent activation of mPFC-PVIs"

- Lines 286-288: "sEPSCs and sIPSCs showed a developmental decrease in frequency as well as a significant main effect of age housing, showing a decrease in sEPSC frequency between jSI and GH (Fig. 5c, 288 left: 2-way ANOVA, age factor: $p < 0.001$, housing factor: $p = 0.01$)."

- Line 350: "shared across of a range of disorders"

- Line 421: "social isolation for two weeks does not impact does not impact adult social behavior"

- Line 743: "reveals significant differences between the joint distribution matrix between of jSI and GH mice"

- Line 752: "Change in z-score from baseline in the first active 753 encounter shows a significant difference"

- Line 895...: where is the mouse placed at the beginning of the Light/dark test ?

- Line 929-931: "To compare social vs. object mPFC-PVI activity, we compared mean z-score during baseline periods (30 sec before introducing a stimulus) compared with the 30 sec after introducing either a 931 social stimulus or an object."

- Line 943-44: "If mice stayed in the center, Chamber entries (social chamber entry, object chamber entry, or remain in center chamber) within 5 seconds of the end of the pulse...": Sentence not very understandable sentence. If the mouse stays in the center, we cannot measure entries...!

- Line 975: anterior cingulate cortex

Response: *Thank you, we have edited these typos and increased clarity of these sentences, reflected in blue text in the revised manuscript.*

REVIEWERS' COMMENTS:

Reviewer #1 (Remarks to the Author):

In my view, the authors have done a very nice job of responding to my critique, as well as the issues raised by the other reviewers. The addition of substantial new data was much appreciated, especially the new analyses examining behavior transitions, and the new data in Fig. S14. The revised manuscript will be of interest to this journal's general audience, and I feel that it's suitable for publication in its present form.

Reviewer #2 (Remarks to the Author):

The authors have satisfactorily responded to all questions and made the necessary changes to the manuscript.

Reviewer #3 (Remarks to the Author):

The authors have convincingly answered to my comments

Response to Reviewers

Dear Reviewers:

We are grateful for your time and thoughtful comments to our manuscript.

Hirofumi Morishita MDPHD (on behalf of all the authors)

REVIEWERS' COMMENTS:

Reviewer #1 (Remarks to the Author):

In my view, the authors have done a very nice job of responding to my critique, as well as the issues raised by the other reviewers. The addition of substantial new data was much appreciated, especially the new analyses examining behavior transitions, and the new data in Fig. S14. The revised manuscript will be of interest to this journal's general audience, and I feel that it's suitable for publication in its present form.

Reviewer #2 (Remarks to the Author):

The authors have satisfactorily responded to all questions and made the necessary changes to the manuscript.

Reviewer #3 (Remarks to the Author):

The authors have convincingly answered to my comments